# Novel Oxadiazole-Quinoxalines as Hybrid Scaffolds with Antitumor Activity

**DOI:** 10.3390/ijms26041439

**Published:** 2025-02-08

**Authors:** Paola Corona, Stefania Gessi, Roberta Ibba, Stefania Merighi, Prisco Mirandola, Gérard Aimè Pinna, Manuela Nigro, Giulia Pozzi, Battistina Asproni, Alessia Travagli, Sandra Piras, Antonio Carta, Paola Caria, Gabriele Murineddu

**Affiliations:** 1Department of Medicine, Surgery and Pharmacy, University of Sassari, 07100 Sassari, Italy; ribba@uniss.it (R.I.); pinger@uniss.it (G.A.P.); asproni@uniss.it (B.A.); piras@uniss.it (S.P.); acarta@uniss.it (A.C.); muri@uniss.it (G.M.); 2Department of Translational Medicine, University of Ferrara, 44121 Ferrara, Italy; gss@unife.it (S.G.); mhs@unife.it (S.M.); ngrmnl@unife.it (M.N.); trvlss@unife.it (A.T.); 3Department of Medicine and Surgery, University of Parma, Via Gramsci 14, 43126 Parma, Italy; prisco.mirandola@unipr.it (P.M.); giulia.pozzi@unipr.it (G.P.); 4Department of Biomedical Sciences, University of Cagliari, Cittadella Universitaria, 09042 Monserrato, Italy; paola.caria@unica.it

**Keywords:** 1,3,4-oxadiazoles, quinoxalines, antiproliferative activity, cytotoxicity

## Abstract

A small library of 25 novel 1,3,4-oxadiazole-quinoxalines was synthesized and evaluated in vitro for its cytotoxic activity at 10 μM concentration against nine NCI-different cancer cell lines. Among tested compounds, derivatives **24**, **25**, and **26** showed good inhibition percentages over different cell lines and, therefore, progressed to the full five-dose assay. Compound **24**, possessing a 1,3,4-oxadiazole-core, bearing a 7-trifluoromethyl-quinoxaline nucleus on C-2 and a C-5 phenyl ring, had activity against leukemia, CNS, ovarian, renal, prostate, and breast cancer, with highest the values against breast MCF7 (GI_50_: 1.85 μM) and MDA-MB-468 (GI_50_: 1.95 μM) cell lines, showing the better MG_MID value (−5.02). These novel derivatives were able to delay the S phase of the cell cycle and induce apoptosis.

## 1. Introduction

The constant growing incidence of tumors requires the search for new pharmacological approaches in anticancer therapy that still present problems of selectivity against cancer cells and toxicity against healthy cells. Therefore, the discovery of molecules possessing chemical structures with antitumor activity involves the design of compounds containing functional groups able to interact with specific targets.

Heterocyclic compounds are commonly expressed in molecules with therapeutic activity due to their intrinsic versatility and unique chemical–physical properties [1,2]. About 60% of small molecules with anticancer activity contain a nitrogen-based heterocycle, and an average of approximately 3.2 nitrogen atoms are present in each anticancer drug. The dynamic structure of heterocycles, particularly those that are nitrogen-based, is fundamental both for the size of the ring and for its aromaticity, which contribute to their chemical stability and to the improvement of the activity as it promotes stable bonds with targets, determining a characteristic mechanism of action of the drug. Over the last few decades, the design of new oxadiazole or quinoxaline-based drugs in medicinal chemistry has been considerably accelerated, many of these passing to preclinical phase studies or to the market for their different therapeutic potentials, for example, as anticancer drugs. In fact, among compounds that possess antitumor activity, about 8% have one or more oxygen-based heterocycles in their structure as quinoxaline [3,4,5] and 1,3,4-oxadiazole [6,7,8] rings, which play a pivotal role as widely reported. The interesting results obtained could be attributed, at least in part, to the following features: the two nuclei, 1,3,4-oxadiazole and quinoxaline, are versatile precursors that also promote a certain robustness and chemical stability, also fostering the insertion of different functional groups both lipophilic and hydrophilic. The different orientation of these areas of modification may lead to the bonding of functionalizing residues and to their optimal spatial conformation. The groups in these ring systems involving nitrogen and oxygen atoms can act as donors or acceptors of electrons, thus, improving the interactions.

Our identification of a series of 1,3,4-oxadiazoles derivatives of general structure **1** (Figure 1) with high toxicity versus HeLa and MCF-7 cell lines on one side [9,10] and the anticancer activity of 2,3-dichloroquinoxaline-based compounds [11] (**2**, Figure 1) on the other, prompted us to further investigate the activity of such interesting heterocycles.

1,3,4-Oxadiazoles and quinoxalines must possess several structural features to be used as potential anticancer agents. Firstly, the 1,3,4-oxadiazole isomer, which is more active and, therefore, preferred, allows a great variety of substituents in the two free positions, as the introduction of an aromatic portion greatly increases the pharmacological effect of the compound [12]; for this reason, conjugation with a quinoxaline nucleus suggests a good synthetic approach, as the combination of two rings that are already biologically active could determine their synergism with consequent enhancement of their common action. Moreover, a large nitrogen area mimics many endogenous physiological molecules, which could confer a lower toxicity against non-malignant cells.

Secondly, the quinoxaline showed the best activity when conjugated with nitrogen or oxygenated heterocyclic systems, especially when it carries a chlorine atom in position C3. Even in the case of this bicyclic nucleus, the best potency is observed when it carries two substituents in positions C2 and C3.

As mentioned, various anticancer drugs containing these heterocyclic nuclei have shown, in clinical trials, good potency against several cancers, and this prompted further research on these heterocyclic rings, not only as such but also in combination with each other or with other rings. In this context, structural hybridization [13] has emerged as a promising tool to develop new safe drugs and highly efficient molecules, through possible multi-target interactions, against various diseases, including cancer. A hybrid pharmacophore approach has been adopted to generate combinations of many drugs, designed with the aim to improve bioavailability, alleviate toxicity, and circumvent drug resistances.

Based on the above and in light of our continuing interest on 1,3,4-oxadiazole [9,10] and quinoxaline [14] derivatives endowed with antitumor activity, we postulated that the introduction of such pharmacophores, through a hybrid pharmacophore approach [15], might provide heterocyclic systems with improved activity. Therefore, we designed a quinoxaline-1,3,4-oxadiazole hybrid, which could lead to compounds of general structure **3** (Figure 1) endowed with both interesting antitumor activity and overcome drawbacks such as low solubility, multidrug resistance, and adverse effects [16].

Therein, we report the synthesis and preliminary structure–activity relationship studies of a small library of 25 new hybrid derivatives (**4**–**28**) bearing different substituents both on the phenyl ring and on the quinoxaline heterocycle, shown in Table 1.

## 2. Results and Discussion

### 2.1. Synthesis of 1,3,4-oxadiazole-chloroquinoxalines

The 1,3,4-oxadiazole-chloroqinoxalines **4**–**26** were synthesized as reported in Figure 1. The reaction of appropriate *o*-phenylendiamine **29**–**32** with diethylketomalonate in refluxing ethanol gave compounds **33**–**39** as described in the literature [17,18,19,20]. After flash chromatography separations of isomers **34**/**35**, **36**/**37**, and **38**/**39** with appropriate solvent mixtures, as below reported in the Section 3, the esters **33**–**36**, **38**, and **39**, but not isomer **37** isolated in very low yield, were reacted with hydrate hydrazine 98% in refluxing ethanol to give the corresponding hydrazides **40**–**45** in quantitative yields. The *N*-acylation with the appropriate acyl chloride, using *N*-methylpyrrolidone as the solvent, yielded the corresponding acyl hydrazides **46**–**68** whose treatment with POCl_3_ at 110 °C gave the chloroquinoxaline derivatives **4**–**26**.

Unequivocal structures to the pairs of isomeric esters **34** (6-OCH_3_) and **35** (7-OCH_3_), **36** (6-Cl) and **37** (7-Cl), and **38** (6-CF_3_) and **39** (7-CF_3_) (Figure 1) were confirmed by two-dimensional NOE spectroscopy (NOESY) (Appendix A), due to the proximity of C_5_-H and N_4_-H atom in the structures. In the 6-substituted esters (**34**, **36** and **38**), the doublet relative to C_5_-H have a coupling constant of 1–3 Hz (J *meta*), due to the peak of correlation with the proton bounded to quinoxaline nitrogen N_4_, whereas for 7-substituted isomers (**35**, **37**, and **39**), the same correlation give a doublet with a coupling constant from 6 to 10 Hz (J *ortho*).

The chloro-quinoxalines **27** and **28** were prepared by a synthetic route (Figure 2), as reported previously for compounds **4**–**26**. The four-step synthetic route started from *o*-phenylendiamine **29** whose reaction with dimethylacetylendicarboxilate in methanol at r.t. gave the mixture of four *endo*/*exo* and *cis*/*trans* isomers of compounds **69** [21] that reacted with hydrate hydrazine 98% and furnished the hydrazides mixture **70**. The reaction of **70** with benzoyl chloride or 4-Cl-benzoyl chloride led to intermediates **71** and **72**, respectively, as isomers *E* only. Their cyclization in refluxing POCl_3_ yielded the desired compounds **27** and **28**.

### 2.2. Biology

All title compounds **4**–**28** were submitted to the National Cancer Institute (NCI, USA) for the human tumor cell screen and were tested initially at a single high dose (10^−5^ M) in the full NCI 60 cell panel (a one-dose graph is provided in the Appendix A). Those compounds that satisfy pre-determined threshold inhibition criteria in a minimum number of cell lines will progress to the full five-dose assay [22].

Nine compounds, **4**, **8**, **9**, **10**, **11**, **18**, **19**, **20**, and **23** (data reported in the Appendix A) showed interesting activity against nine different cancer cell lines. All derivatives were active against breast cancer cell lines (Table 2) with percentage growth inhibition ranging from 40.65% to 92.89%, also showing interesting cytotoxicity values for compound **8** in MCF-7 and T-47D cells and compounds **18** and **23** in MDA-MB-468.

Among them, derivatives **8**, **18**, and **19** elicited significant values of percentage growth inhibition against different cell lines of cancer. Particularly, compound **8** displayed percentage growth inhibition over 85% against three different cancer cell lines, HCT-116 (colon cancer, 85.83%), HL-60(TB) (leukemia, 86.43%), and HOP-62 (non-small cell lung cancer, 88.89%).

Compounds **18** and **19** exhibited moderate-to-good activity against leukemia HL-60(TB) cell lines, 43.43% and 60.93%, respectively. Interestingly, compound **18** was accompanied with cytotoxicity against the ovarian cancer OVCAR-3 cell line, whereas only compound **19** resulted cytotoxic against renal cancer ACHN and UO-31 cell lines. In addition, derivative **23** provided to be active against the five leukemia cell lines (Table 3) and was able to evoke percentage growth inhibition for cell lines NCI-H522M (non-small cell lung cancer), HCT-116, and HCT-15 (colon cancer) with values of 61.14%, 41.44%, and 69.90%, respectively.

In this small library of novel quinoxaline-oxadiazoles, three compounds, **24**, **25**, and **26** were able to inhibit the cellular growth in all panel tumor cell lines, except for the prostate cancer panel for compound **25**, resulting the most interesting among synthesized derivatives based on their preliminary anticancer assay. Moreover, they resulted cytotoxic against several cancer cell lines (Appendix A).

Briefly, compounds **24**–**26** resulted cytotoxic against all leukemia cell lines with percentage values ranging from 4.01% (**24**, CCRF-CEM) to 57.94% (**26**, HL-60(TB)), except for compound **25**, showing a value of 99.36% growth inhibition against the CCRF-CEM cell line.

Concerning the nine cell lines of non-small cell lung cancer, derivatives **24**–**26** exhibited cytotoxicity against the HOP-92 cell line, and only compounds **25** and **26** elicited a cytotoxic activity against NCI-H23 and NCI-H522 cell lines, whereas analog **24** also inhibited the NCI-H322M cell line.

High cytotoxicity values were elicited from cell lines HCT-116 and SW-620 (84.26–87.08%) of the colon cancer panel, and there was significant percentage growth inhibition against the KM12 cell line (77.86% for compound **24**).

Among the cell lines for CNS cancer, compounds **24**–**26** showed cytotoxicity values against SF-268 and SF-539 cell lines. Moreover, a very interesting value of growth inhibition (81.21%) was observed for compound **24** against the SF-268 cell line.

Regarding the melanoma panel, compound **26** represents the term with the highest GI% value (91.85%, SK-MEL-2), whereas **25** has the highest cytotoxic against MALME-3M and SK-MEL-5 cell lines. Likewise, derivatives **24** and **26** provided high cytotoxicity values against LOX IMVI and SK-MEL-28 cell lines, respectively.

Compounds **24**–**26** had very high cytotoxicity against the OVCAR-3 cell line with values ranging from 71.24% to 93.14%. The biological profile of the three derivatives was in line with those of the other panels except for OVCAR-5 and SK-OV-3 cell lines whose percentage growth inhibition values were <40%.

Compounds **24**–**26** were also active against RXF 393 and SN12C cell lines of the renal cancer panel. Compound **26** showed interesting GI% values for ACHN, CAKI-1, and UO-31 cell lines with respect to analogs **24** and **25**, which had no significant (<40%) GI% values.

Regarding prostate cancer, compounds **24** and **26** provided values of GI% > 84.81% against PC-3 and DU-145 cell lines.

Finally, the one-dose (10 µM) assay of the three compounds in the six cell lines of breast cancer highlighted the most important data, showing GI% values ranging from 57.49% to 94.82%, whereas their cytotoxicity values range from 2.05% to 85.67%.

Particularly, compound **24** gave high percentages of growth inhibition against BT-549 and HS 578T cell lines. On the other hand, derivative **25** resulted cytotoxic against all the six cell lines. Moreover, analog **26** had a biological profile like **25** with high level of cytotoxicity and very good GI% (87.77%) against the T-47D cell line.

Among the breast cancer panel, the BT-549 cell line represents the main target of these three compounds as derivative **24** inhibits it for 94.82%, and **25** and **26** have a cytotoxicity of 85.67% and 80.67%, respectively.

In the light of these results, among the twenty-five compounds synthesized, derivatives **24**, **25**, and **26** satisfied a pre-determined threshold of inhibition criteria, passing to the five-dose screen against a panel of about 60 different tumor cell lines [23,24,25,26,27,28,29,30,31].

Table 4 reports the GI_50_ values of these compounds in the five-dose screen for each cell line (Appendix A). Furthermore, a mean graph midpoint (MG_MD) has been calculated [32,33] (Appendix A).

Compound **24** exhibited high activity against leukemia cell lines (GI_50_: 3.01 μM–4.21 μM), single cell lines of other tumors, and the five breast cancer cell lines (GI_50_: 1.85 μM–4.33 μM) as shown in Table 4. Concerning the sensitivity against each cell line, compound **24** showed the highest activity against the breast MCF7 cell line (GI_50_: 1.85 μM) and least against the non-small lung cancer A549/ATCC cell line (GI_50_: 18.6 μM).

Derivatives **25** and **26** showed an anticancer activity against few of the tested cell lines: both were active against five different leukemia cancer, but only compound **26** was also active against melanoma cancer LOX IMVI (GI_50_: 3.31 μM) and two cell lines of breast cancer: MCF7 (GI_50_: 3.31 μM) and T-47D (GI_50_: 4.48 μM).

In Table 5, we reported the average of Log_10_ GI_50_ for each cancer panel to underline both the activity and selectivity for compounds **24–26**. All derivatives showed selectivity against leukemia cell lines. Among them, the best result was obtained for compound **24** with a Log_10_ GI_50_ value of −5.34 (4.57 μM). Moreover, compounds **24** and **26** were selective against breast cancer with Log_10_ GI_50_ values of −5.45 (3.55 μM) and −5.12 (7.58 μM), respectively. Finally, compound **24** showed selectivity against prostate cancer with a Log_10_ GI_50_ value of −5.24 (5.75 μM), recording an MG_MID log_10_ GI_50_ = −5.02, which accounts for the highest micromolar activity against all cell lines.

In conclusion, the three compounds are safe and not lethal against leukemia cell lines according to the LC_50_ values reported in Table 6.

Derivates **5**, **6**, **7**, **12**, **13**, **14**, **15**, **16**, **17**, **21**, **22**, **27**, and **28** (data reported in Appendix A) showed no significant growth percentage inhibition values (<40%).

#### 2.2.1. MTS Assay

In the light of the cytotoxicity of these novel derivatives, particularly against leukemia, we further evaluate their activity against new different human tumor cell lines such as NB4 (acute promyelocytic leukemia) and JURKAT (human T lymphocytic cells) using a colorimetric test (MTS) for cell proliferation to assess their viability and cytotoxic effect. Among the 25 compounds, we chose the candidates as follows: five derivatives that were not particularly active in the preliminary assays on the 60 cell lines (**5**, **6**, **13**, **14**, and **22**), six that showed activity (**4**, **8**, **9**, **10**, **11**, and **23**), and the three more active compounds **24**, **25**, and **26**. NB4 cells were treated with the novel compounds and incubated for 24, 48, and 72 h. 5-Fluorouracil (5-FU), 10 μM and 100 μM, was used as positive control (Figure 2). The percentage of viable cells were calculated by comparing treated and control cells. The results for NB4 cells show that derivatives **8**, **14**, **24, 25**, and **26** demonstrated a good antiproliferative activity starting at 24 h and continuing through 48 and 72 h. However, compound **25** ceases its cytotoxic effects at 48 h. Compounds **4**, **6**, and **9** show a significant cytotoxic effect starting from 48 h of treatment until 72 h. In the end, compounds **10** and **23** had a good antiproliferative effect only after 48 h of treatment (64.0% and 62.3%, respectively), whilst the effects of compound **13** were evident only after 72 h of treatment (40.0%) (Figure 2 and Appendix A).

Consequently, the most active compounds **4**, **8**, **9**, **24**, **25**, and **26**, were tested at lower concentrations, 100 nM and 1 μM, for 72 h (Figure 3 and Appendix A). These latter results suggest that all the evaluated compounds did not provide significant cytotoxic activity in the range of doses 100 nM–1 μM on NB4 leukemia cells.

The same compounds were tested at 10 μM against the JURKAT cell line (Figure 4, Appendix A), within which derivative **23** showed the highest cytotoxicity values (73.0% after 24 h, 74.0% after 48 h, and 69.3% after 72 h). Analogs **10**, **24**, **25**, and **26** were particularly active only at 24 and 48 h, with cytotoxicity values ranging from 41.0% to 68.7%. Compounds **9** and **11** showed cytotoxicity at all three treatment times, exceeding 50% values after 48 h and 72 h. Derivatives **5** and **22** showed modest but significant cytotoxicity only after 24 h of treatment.

Furthermore, the fourteen selected compounds were evaluated against SH-SY5Y, A375, and MAHLAVU cell lines. As for SH-SY5Y cells, compounds **23**, **24, 25**, and **26** showed significant cytotoxicity values ranging from 35.8% to 44.6% at 24 h (Figure 5, Appendix A). In the A375 cell line, cytotoxicity values of all compounds investigated are less than 50%, except for derivative **24**, which showed cytotoxicity values of 60.3% (24 h), 73.7% (48 h), and 56.7% (72 h) (Figure 6, Appendix A). No significant cytotoxic effects were observed in MAHLAVU cancer cells except for compound **24**, which showed small but significant cytotoxicity from 24 h (16.8%) to 72 h (32.8%) (Figure 7, Appendix A).

Finally, we evaluated the effect of novel compounds, measuring the cell survival of human lymphocytes (Figure 8, Appendix A), used as a model of non-tumoral cells. After 24 h, ten compounds (**4**, **5**, **6**, **8**, **9**, **10**, **13**, **14**, **22**, and **23**) showed no cytotoxicity, while derivatives **11**, **24**, **25**, and **26** had slight cytotoxicity of 19.3%, 33.4%, 26.5%, and 24.8%, respectively.

#### 2.2.2. Flow Cytometry Assay

The effects of compounds **24**, **25**, and **26** on cell cycle and cell viability were evaluated by flow cytometry in the two tumorigenic cell lines: JURKAT and SH-SY5Y. Each cell line was treated with compounds **24**, **25**, and **26** (10 µM) or 5-fluorouracil (5-FU, 10 µM), used as a positive control, and compared to cells treated with DMSO (negative control cells, CTRL). For the cell viability analysis, we evaluated the percentage of viable cells (% of Annexin V^−^/7-AAD^−^ cells), cells in early apoptosis (% of Annexin V^+^/7-AAD^−^ cells), and cells in late apoptosis/necrosis (% of Annexin V^+^/7-AAD^+^ cells) after 6 h of treatment.

The results obtained in JURKAT and SH-SY5Y cells just after 6 h of treatment are shown in Figure 9. All compounds tested in JURKAT cells induced a drastic reduction in cell viability: compounds **24** and **25** mainly via an apoptotic mechanism, while compound **26** induced mainly necrosis (Figure 9A,C). Indeed, the percentage of early apoptotic cells, identified as cells positive for Annexin V but negative for 7-AAD, was significantly higher in cells treated with compounds **24** and **25** as compared to both the untreated cells and the cells treated with 5-FU (**, §§, *p* < 0.01 vs. CTRL and 5-FU, respectively). By contrast, in cells treated with compound **26**, we observed an extremely elevated percentage of necrotic cells identified as positive for both Annexin V and 7-AAD. Therefore, compound **26** showed a strong cytotoxic effect with the highest percentage of necrotic cells as compared to control cells and cells treated with all other compounds (***, §§§, ###, $$$, *p* < 0.001 vs. CTRL, 5-FU, **24** and **25**).

Concerning SH-SY-5Y cells, compounds **25** and **26** exhibited the highest levels of cytotoxicity, inducing a significant reduction in cell viability mainly through necrosis (Figure 9B,D). However, we found a significant percentage of apoptotic cells in those treated with compounds **25** and **26,** compared to CTRL, 5-FU, and compound **24** (**, §§, ##, *p* < 0.01 vs. CTRL, 5-FU, 24). In SH-SY-5Y, no significant cytotoxic effects were observed in cells treated with compound **24**.

After 24 h of treatment, we evaluated whether the compounds could affect the cell cycle, by assessing the percentage of cells in the G1, S, and G2/M phases (Figure 10). In both tumorigenic cell lines, JURKAT and SH-SY5Y, treated with 5-FU, we observed a significant increase in the percentage of cells blocked in the S phase. These results suggest that 5-FU may slow the cell cycle rate without completely inhibiting DNA replication (Figure 10A,B).

Compounds **25** and **26** exhibited detrimental effects on the cell cycle. Indeed, we found that both JURKAT and SH-SY5Y cells treated with compound **26** were significantly enriched in cells blocked in the S phase as compared to control cells and cells treated with compound **24** (Figure 10A,B: *, *p* < 0.05 vs. CTRL and ##, *p* < 0.1 vs. **24** in JURKAT; ***, ### *p* < 0.001 vs. CTRL and **24**, in SH-SY5Y), with a consequent reduction in the percentage of cells in the G1 phase. The effects of compound **26** in these cell lines were comparable to 5-FU. Concerning compound **25**, it induced a significant increase in the percentage of cells in the S phase only in SH-SY5Y cells (*, # *p* < 0.05 vs. CTRL and **24**, respectively) but with a milder effect than 5-FU. Moreover, compounds **25** and **26** induced significant DNA fragmentation after 24 h of treatment (Figure 10C,D).

The effects of compounds on cell cycle and cell viability were evaluated by flow cytometry in normal non-tumorigenic human lymphocytes. After 24 h of treatment, compounds **24** and **25** induced DNA fragmentation, indicating apoptosis in activated lymphocytes. However, only compound **24** had a significant effect on the cell cycle, increasing the percentage of cells in both the S phase and the G2/M phase (Table 7). Interestingly, compound **26** did not affect either apoptosis or the cell cycle.

## 3. Methods and Materials

### 3.1. Chemistry

#### 3.1.1. General Methods

All reactions involving air or moisture-sensitive compounds were performed with atmospheric N_2_. Solvents and reagents were obtained from commercial suppliers and were used without further purification. Quinoxalinone-esters **33** [17], **34** and **35** [18], **36** and **37** [19], **38** and **39** [20], and **69** [21] were synthesized and identified according to and by comparison with literature.

Flash column chromatography was performed automatically on Flash-master (Biotage^®^, Uppsala, Sweden) with pre-packed Biotage^®^ SNAP silica gel cartridges or manually on silica gel (Kieselgel 60, 0.040−0.063 mm, Merck^®^, Darmstadt, Germany). Thin-layer chromatography (TLC) was performed with Polygram SIL N-HR/HV_254_ pre-coated plastic sheets (0.2 mm) on aluminum sheets (Kieselgel 60 F254, Merck^®^). Melting points were obtained on a Köfler melting point apparatus and are uncorrected.

IR spectra were recorded as KBr pellet with a Jasco^®^ (Cremella, Italy) FT/IR 460 plus spectrophotometer and are expressed in ν (cm^−1^). NMR experiments were recorded at room temperature and were run on a Bruker^®^ (Fallanden, Switzerland) Avance III Nanoboy 400 system (400.13 MHz for ^1^H, and 100.62 MHz for ^13^C). Spectra were acquired using deuterated chloroform (chloroform-d) or deuterated dimethylsulfoxide (DMSO-*d*_6_) as solvents. Chemical shifts (δ) for ^1^H- and ^13^C-NMR spectra are reported in parts per million (ppm) using the residual non-deuterated solvent resonance as the internal standard (for chloroform-δ: 7.26 ppm, ^1^H and 77.16 ppm, ^13^C; for DMSO-d_6_: 2.50 ppm, ^1^H, 39.52 ppm, ^13^C). Data are reported as follows: chemical shift (sorted in descending order), multiplicity (s for singlet, br s for broad singlet, d for doublet, dd for double doublet, dt for double triplet, t for triplet, and m for multiplet), and integration and coupling constants (J) in Hertz (Hz) (Appendix A). Spectroscopic data are in accordance with the structures. LC/MS analyses were run on an Agilent^®^ (Santa Clara, CA, USA) 1100 LC/MSD system consisting of a single quadrupole detector (SQD) mass spectrometer (MS) equipped with an electrospray ionization (ESI) interface and a photodiode array (PDA) detector, range 120−550 nm. ESI in positive mode was applied. Mobile phases: (A) MeOH in H_2_O (8:2). Analyses were performed with a flow rate of 0.9 mL/min and temperature of 350 °C. The purity of all final compounds was determined by elemental analysis on a PerkinElmer^®^ (Waltham, MA, USA) 240B analyzer (C, H, and N). All the final compounds were found to be >95% when analyzed.

#### 3.1.2. General Procedure I: Synthesis of Hydrazides **40**–**45** and **70**

To a solution of ester **33–39** (5 mmol) in absolute EtOH (2 mL), hydrazine monohydrate 98% (10 mmol, 2 eq) was added and the reaction was refluxed for 2 h. After cooling at room temperature, the resulting precipitate was filtered, washed (H_2_O), and air-dried to obtain the desired hydrazide as solid.

#### 3.1.3. 3-Oxo-3,4-dihydroquinoxaline-2-carbohydrazide (**40**)

The title compound was prepared from **33** [17] using the general procedure I to afford **40** as a yellow solid (0.095 g, 93%), after trituration with petroleum ether, mp > 270 °C (dec). ^1^H NMR (400 MHz, DMSO-*d*_6_) *δ*: 4.69 (br s, 2H, NH_2_ exch. with D_2_O), 7.35–7.40 (m, 2H), 7.63 (t, 1H, J_o_ = 8.4 Hz, J_m_ = 1.2 Hz), 7.85 (dd, 1H, J_o_ = 8.4 Hz, J_m_ = 1.2 Hz), 10.06 (br s, 1H, NH exch. with D_2_O), 12.97 (br s, 1H, NH exch. with D_2_O).

#### 3.1.4. 6-Methoxy-3-oxo-3,4-dihydroquinoxaline-2-carbohydrazide (**41**)

The title compound was prepared from **34** [18] using the general procedure I to afford **41** as a brown solid (0.116 g, 99%), mp > 280 °C (dec). ^1^H NMR (400 MHz, DMSO-*d*_6_) *δ*: 3.86 (s, 3H), 4.68 (br s, 2H, NH_2_ exch. with D_2_O), 6.79 (s, 1H), 6.98 (d, 1H, J = 8.0 Hz), 7.77 (d, 1H, J = 8.8 Hz), 9.13 (br s, 1H, NH exch. with D_2_O), 10.47 (br s, 1H, NH exch. with D_2_O).

#### 3.1.5. 7-Methoxy-3-oxo-3,4-dihydroquinoxaline-2-carbohydrazide (**42**)

The title compound was prepared from **35** [18] using the general procedure I to afford **42** as a yellow solid (0.109 g, 93%), mp > 280 °C (dec). ^1^H NMR (400 MHz, DMSO-*d*_6_) *δ*: 3.84 (s, 3H), 4.20 (br s, 1H, NH_2_ exch. with D_2_O), 7.22–7.37 (m, 2H), 7.76–7.78 (m, 1H), 9.18 (br s, 1H, NH exch. with D_2_O), 10.30 (br s, 1H, NH exch. with D_2_O).

#### 3.1.6. 6-Chloro-3-oxo-3,4-dihydroquinoxaline-2-carbohydrazide (**43**)

The title compound was prepared from **36** [19] using the general procedure I to afford **43** as a yellow solid (0.051 g, 43%), mp > 300 °C (dec). ^1^H NMR (400 MHz, DMSO-*d*_6_) *δ*: 3.44 (br s, 2H, NH_2_ exch. with D_2_O), 7.41 (d, 1H, J = 8.0 Hz), 7.71 (d, 1H, J = 8.4 Hz), 7.95 (s, 1H), 9.28 (br s, 1H, NH exch. with D_2_O), 10.96 (br s, 1H, NH exch. with D_2_O).

#### 3.1.7. 3-Oxo-6-trifluoromethyl-3,4-dihydroquinoxaline-2-carbohydrazide (**44**)

The title compound was prepared from **38** [20] using the general procedure I to afford **44** as a yellow solid (0.074 g, 55%), mp > 360 °C (dec). ^1^H NMR (400 MHz, DMSO-*d*_6_) *δ*: 5.07 (br s, 2H, NH_2_ exch. with D_2_O), 7.50 (dd, 1H, J_o_ = 8.4 Hz, J_m_ = 1.2 Hz), 7.67 (s, 1H), 8.00 (d, 1H, J = 8.4 Hz), 9.32 (br s, 1H, NH exch. with D_2_O), 11.80 (br s, 1H, NH exch. with D_2_O).

#### 3.1.8. 3-Oxo-7-trifluoromethyl-3,4-dihydroquinoxaline-2-carbohydrazide (**45**)

The title compound was prepared from **39** [20] using the general procedure I to afford **45** as a yellow solid (0.095 g, 90%), mp = 287–289 °C. ^1^H NMR (400 MHz, DMSO-*d*_6_) *δ*: 5.43 (br s, 2H, NH_2_ exch. with D_2_O); 7.44 (dd, 1H, J = 8.8 Hz), 7.69 (d, 1H), 8.06 (s, 1H), 9.35 (s, 1H, NH exch. with D_2_O), 12.02 (br s, 1H, NH exch. with D_2_O).

#### 3.1.9. (*E*)-2-(3-Oxo-3,4-dihydroquinoxalin-2(1H)-ylidene)acetohydrazide (**70**)

To a solution of ester **69** [21] (3 mmol) in absolute EtOH (20 mL), hydrazine monohydrate 98% (6 mmol, 2 eq) was added and the reaction was refluxed for 3 h and then poured into ice. The resulting precipitate was filtered, washed (H_2_O), and air-dried to obtain title compound **70** as a white solid (0.049 g, 73.02%), mp = 237–239 °C. ^1^H-NMR δ (DMSO-*d*_6_): 4.28 (br s, 2H, NH_2_ exch. with D_2_O), 5.59 (s, 1H), 6.89 (t, 1H, J = 8.0 Hz), 6.96 (t, 1H, J = 8.0 Hz), 7.10 (d, 1H, J = 7.6 Hz), 7.29 (d, 1H, J = 7.6 Hz), 9.14 (br s, 2H, NH × 2 exch. with D_2_O), 11.62 (br s, 1H, NH exch. with D_2_O).

#### 3.1.10. General Method II: Synthesis of Acylhydrazides **46**–**68** and **71**,**72**

To a solution of **40** or **41***–***45** (0.730 mmol), *N*-methyl-2-pyrrolidone (3.1 mL) was added dropwise the required acyl chloride (0.730 mmol, 1.0 eq). After stirring the reaction for 24 h at room temperature, the mixture was poured into ice/water. The precipitate was filtered, washed with water, and dried to afford the corresponding products **46–68**.

#### 3.1.11. N′-Benzoyl-3-oxo-3,4-dihydroquinoxaline-2-carboidrazide (**46**)

The title compound was prepared from **40** and benzoyl chloride using the general procedure II to afford **46** as a yellow solid (0.128 g, 57%), mp > 280 °C (dec). IR (cm^−1^): 1701.48 (C=O), 3419 (NH). ^1^H NMR (400 MHz, DMSO-*d*_6_) *δ*: 7.42 (t, 2H, J = 8.0 Hz), 7.53 (t, 2H, J = 7.6 Hz), 7.61 (t, 1H, J = 7.6 Hz), 7.68 (t, 1H, J = 8.0 Hz), 7.91–7.95 (m, 3H), 10.91 (s, 1H, NH exch. with D_2_O), 11.09 (s, 1H, NH exch. with D_2_O), 12.95 (br s, 1H, NH exch. with D_2_O).

#### 3.1.12. N′-(4-Methylbenzoyl)-3-oxo-3,4-dihydroquinoxaline-2-carbohydrazide (**47**)

The title compound was prepared from **40** and 4-methylbenzoyl chloride using the general procedure II to afford **47** as a white solid (0.221 g, 94%), mp = 256 °C. IR (cm^−1^): 1619.51 (C=O), 1655.16 (C=O), 1698.04 (C=O), 3447.82 (NH). ^1^H NMR (400 MHz, DMSO-*d*_6_) *δ*: 2.39 (s, 3H), 7.33 (d, 2H, J_o_ = 8.0 Hz), 7.39–7.44 (m, 2H), 7.66 (t, 1H, J_o_ = 8.0 Hz), 7.86 (d, 2H, J_o_ = 8.0 Hz), 7.92 (d, 1H, J_o_ = 8.0 Hz), 10.84 (d, 1H, J = 1.6 Hz, NH exch. with D_2_O), 11.70 (d, 1H, J = 1.6 Hz, NH exch. with D_2_O), 12.95 (s, 1H, NH exch. with D_2_O).

#### 3.1.13. N′-(4-Methoxybenzoyl)-3-oxo-3,4-dihydroquinoxaline-2-carbohydrazide (**48**)

The title compound was prepared from **40** and 4-methoxybenzoyl chloride using the general procedure II to afford **48** as a brown solid (0.106 g, 43%), mp > 250 °C (dec). IR (cm^−1^): 1612.10 (C=O), 1655.00 (C=O), 1683.29 (C=O), 3440.10 (NH). ^1^H NMR (400 MHz, DMSO-*d*_6_) *δ*: 3.84 (s, 3H), 7.06 (d, 2H, J_o_ = 8.8 Hz), 7.39–7.43 (m, 2H), 7.67 (t, 1H, J_o_ = 7.6 Hz), 7.91–7.85 (m, 1H), 7.94 (d, 2H, J_o_ = 8.8 Hz), 10.77 (s, 1H, NH, exch. with D_2_O), 11.09 (s, 1H, NH, exch. with D_2_O), 12.92 (br s, 1H, NH exch. with D_2_O).

#### 3.1.14. N′-4-Chlorobenzoyl-3-oxo-3,4-dihydroquinoxaline-2-carbohydrazide (**49**)

The title compound was prepared from **40** and 4-chlorobenzoyl chloride using the general procedure II to afford **49** as a yellow solid (0.235 g, 94%), mp > 280 °C (dec). IR (cm^−1^): 1698.04 (C=O), 1619.51 (C=O), 3447.82 (NH). ^1^H NMR (400 MHz, DMSO-*d*_6_) *δ*: 7.42 (t, 1H, J = 8.0 Hz), 7.57 (d, 2H, J = 8.4 Hz), 7.60–7.62 (m, 1H), 7.68 (t, 1H, J = 8.0 Hz), 7.91–7.97 (m, 3H), 11.04 (s, 1H, NH exch. with D_2_O), 11.12 (s, 1H, J = 1.6 Hz, NH exch. with D_2_O), 12.95 (br s, 1H, NH exch. with D_2_O).

#### 3.1.15. N′-(4-(*tert*-Butyl)benzoyl-3-oxo-3,4-dihydroquinoxaline-2-carbohydrazide (**50**)

The title compound was prepared from **40** and 4-*tert*-butylbenzoyl chloride using the general procedure II to afford **50** as a yellow solid (0.234 g, 88%), mp > 250 °C (dec). IR (cm^−1^): 1627.07 (C=O), 1655.18 (C=O), 1688.70 (C=O), 3390.59 (NH). ^1^H NMR (400 MHz, DMSO-*d*_6_) *δ*: 1.30 (s, 9H), 7.39–7.43 (m, 2H), 7.54 (d, 2H, J_o_ = 8.0 Hz), 7.67 (s, 1H, J_o_ = 7.2 Hz), 7.88–7.93 (m, 3H), 10.84 (s, 1H, NH exch. with D_2_O), 11.09 (s, 1H, NH exch. with con D_2_O), 12.92 (br s, 1H, NH exch. with D_2_O).

#### 3.1.16. N′-(Cyclohexanecarbonyl)-3-oxo-3,4-dihydroquinoxaline-2-carbohydrazide (**51**)

The title compound was prepared from **40** and cyclohexancarbonyl chloride using the general procedure II to afford **51** as a white solid (0.209 g, 91%), mp > 250 °C (dec). IR (cm^−1^): 1624.60 (C=O), 1664.76 (C=O), 1690.29 (C=O), 3227.84 (NH). ^1^H NMR (400 MHz, DMSO-*d*_6_) *δ*: 1.21–1.29 (m, 3H), 1.36–1.44 (m, 2H), 1.60–1.63 (m, 1H), 1.65–1.75 (m, 4H), 2.30–2.36 (m, 1H), 7.38–7.42 (m, 2H), 7.66 (t, 1H, J_o_ = 7.2 Hz), 7.90 (d, 1H, J_o_ = 8.0 Hz), 10.50 (d, 1H, J = 3.6 Hz, NH exch. with D_2_O), 11.20 (d, 1H, J = 3.6 Hz, NH exch. with D_2_O), 12.96 (s, 1H, NH exch. with D_2_O).

#### 3.1.17. N′-(3-Methylbenzoyl)-3-oxo-3,4-dihydroquinoxaline-2-carbohydrazide (**52**)

The title compound was prepared from **40** and 3-methylbenzoyl chloride using the general procedure II to afford **52** as a yellow solid (0.212 g, 90%), mp > 250 °C (dec). IR (cm^−1^): 1626.56 (C=O), 1664.24 (C=O), 1693.79 (C=O), 3215.68 (NH). ^1^H NMR (400 MHz, DMSO-*d*_6_) *δ*: 2.39 (s, 3H), 7.35–7.45 (m, 4H), 7.67 (t, 1H, J_o_ = 7.6 Hz), 7.74–7.78 (m, 2H), 7.92 (d, 1H, J_o_ = 8.0 Hz), 10.87 (s, 1H, NH exch. with D_2_O), 11.10 (s, 1H, NH exch. with D_2_O), 12.95 (br s, 1H, NH exch. with D_2_O).

#### 3.1.18. N′-(3-Methoxybenzoyl)-3-oxo-3,4-dihydroquinoxaline-2-carbohydrazide (**53**)

The title compound was prepared from **40** and 3-methoxylbenzoyl chloride using the general procedure II to afford **53** as a yellow solid (0.217 g, 88%), mp > 250 °C (dec). IR (cm^−1^): 1622.23 (C=O), 1655.24 (C=O), 1700.01 (C=O), 3186.58 (NH). ^1^H NMR (400 MHz, DMSO-*d*_6_) *δ*: 3.83 (s, 3H), 7.16 (s, 1H), 7.20–7.65 (m, 5H), 7.70 (s, 1H), 7.91 (s, 1H), 10.91 (s, 1H, NH exch. with D_2_O), 11.10 (s, 1H, NH exch. with D_2_O), 12.95 (s, 1H, NH exch. with D_2_O).

#### 3.1.19. 3-Oxo-N′-(3,4,5-trimethoxybenzoyl)-3,4-dihydroquinoxaline-2-carbohydrazide (**54**)

The title compound was prepared from **40** and 3,4,5-trimethoxylbenzoyl chloride using the general procedure II to afford **54** as a yellow solid (0.276 g, 95%), mp > 250 °C (dec). ^1^H NMR (400 MHz, DMSO-*d*_6_) *δ*: 3.74 (s, 3H), 3.86 (s, 6H), 7.31 (s, 2H), 7.35–7.41 (m, 2H), 7.67 (t, 1H, J_o_ = 7.2 Hz), 7.92 (d, 1H, J_o_ = 6.8 Hz), 10.91 (s, 1H, NH exch. with D_2_O), 11.12 (s, 1H, NH exch. with D_2_O), 12.94 (br s, 1H, NH exch. with D_2_O).

#### 3.1.20. N′-2-Chlorobenzoyl-3-oxa-3,4-dihydroquinoxaline-2-carbohydrazide (**55**)

The title compound was prepared from **40** and 2-chlorobenzoyl chloride using the general procedure II to afford **55** as a yellow solid (0.225 g, 90%), mp > 280 °C (dec). ^1^H NMR (400 MHz, DMSO-*d*_6_) *δ*: 7.40–7.57 (m, 6H), 7.68 (t, 1H, J = 7.6 Hz), 7.92 (d, 1H, J = 8.0 Hz), 11.02 (s, 1H, NH exch. with D_2_O), 11.30 (s, 1H, NH exch. with D_2_O) 12.98 (br s, 1H, NH exch. with D_2_O).

#### 3.1.21. N′-3-Chlorobenzoyl-3-oxo-3,4-dihydroquinoxaline-2-carbohydrazide (**56**)

The title compound was prepared from **40** and 3-chlorobenzoyl chloride using the general procedure II to afford **56** as a yellow solid (0.193 g, 77%), mp > 280 °C (dec). ^1^H NMR (400 MHz, DMSO-*d*_6_) *δ*: 7.42 (t, 2H, J = 8.0 Hz), 7.58 (t, 1H, J = 7.6 Hz), 7.68 (t, 2H, J = 8.0 Hz), 7.92 (t, 2H, J = 7.6 Hz), 7.98 (s, 1H), 11.09 (s, 1H, NH exch. with D_2_O), 11.16 (s, 1H, NH exch. with D_2_O), 12.95 (br s, 1H, NH exch. with D_2_O).

#### 3.1.22. N′-2,4-Dichlorobenzoyl-3-oxo-3,4-dihydroquinoxaline-2-carbohydrazide (**57**)

The title compound was prepared from **40** and 2,4-dichlorobenzoyl chloride using the general procedure II to afford **57** as a yellow solid (0.212 g, 77%), mp > 280 °C (dec). ^1^H NMR (400 MHz, DMSO-*d*_6_) *δ*: 7.42 (d, 2H, J = 8.0 Hz), 7.68 (t, 1H, J = 7.6 Hz), 7.83 (d, 1H, J = 8.4 Hz), 7.92 (d, 2H, J = 8.0 Hz), 8.18 (d, 1H, J = 1.0 Hz), 11.18 (s, 1H, NH exch. with D_2_O), 11.20 (s, 1H, NH exch. with D_2_O), 12.96 (br s, 1H, NH exch. with D_2_O).

#### 3.1.23. N′-Benzoyl-6-methoxy-3-oxo-3,4-dihydroquinoxaline-2-carbohydrazide (**58**)

The title compound was prepared from **41** and benzoyl chloride using the general procedure II to afford **58** as a reddish-brown solid (0.158 g, 64%), mp > 280 °C (dec). ^1^H NMR (400 MHz, DMSO-*d*_6_) *δ*: 3.89 (s, 3H), 6.85 (s, 1H), 7.05 (d, 1H, J_o_ = 8.0 Hz), 7.51–7.61 (m, 3H), 7.83 (d, 1H, J_o_ = 8.8 Hz), 7.94 (dd, 2H, J_m_ = 1.2 Hz, J_o_ = 7.2 Hz), 10.93 (s, 1H, NH exch. with D_2_O), 11.18 (s, 1H, NH exch. with D_2_O), 12.93 (br s, 1H, NH exch. with D_2_O).

#### 3.1.24. N′-Benzoyl-7-methoxy-3-oxo-3,4-dihydroquinoxaline-2-carbohydrazide (**59**)

The title compound was prepared from **42** and benzoyl chloride using the general procedure II to afford **59** as a yellow solid (0.222 g, 90%), mp > 280 °C (dec). ^1^H NMR (400 MHz, DMSO-*d*_6_) *δ*: 3.86 (s, 3H), 7.35 (s, 2H), 7.45 (s, 1H), 7.51–7.61 (m, 3H), 7.95 (d, 2H, J_o_ = 7.2 Hz), 10.96 (s, 1H, NH exch. with D_2_O), 11.24 (s, 1H, NH exch. with D_2_O), 12.96 (br s, 1H, NH exch. with D_2_O).

#### 3.1.25. N′-(4-(*tert*-Butyl)benzoyl)-7-methoxy-3-oxo-3,4-dihydroquinoxaline-2-carbohydrazide (**60**)

The title compound was prepared from **42** and 4-*tert*-butylbenzoyl chloride using the general procedure II to afford **60** as a reddish-brown solid (0.224 g, 78%), mp = 295 °C. ^1^H-NMR δ (DMSO-*d_6_*): 1.30 (s, 9H), 3.86 (s, 3H), 7.35 (s, 2H), 7.45 (s, 1H), 7.54 (d, 2H, J = 8.4 Hz), 7.89 (d, 2H, J = 8.0 Hz), 10.88 (s, 1H, NH exch. with D_2_O), 11.23 (s, 1H, NH exch. with D_2_O), 12.96 (s, 1H, NH exch. with D_2_O).

#### 3.1.26. N′-(Cyclohexanecarbonyl)-7-methoxy-3-oxo-3,4-dihydroquinoxaline-2-carbohydrazide (**61**)

The title compound was prepared from **42** and cyclohexanecarbonyl chloride using the general procedure II to afford **61** as a yellow solid (0.234 g, 93%), mp = 333–335 °C. ^1^H-NMR δ (DMSO-*d_6_*): 1.21–1.29 (m, 3H), 1.35–1.44 (m, 2H), 1.63–1.65 (m, 1H), 1.72–1.75 (m, 4H), 2.33 (t, 1H, J = 10.8 Hz), 3.85 (s, 3H), 7.34 (s, 2H), 7.43 (s, 1H), 10.57 (s, 1H, NH exch. with D_2_O), 11.37 (d, 1H, NH exch. with D_2_O, J = 3.2 Hz), 12.96 (s, 1H, NH exch. with D_2_O).

#### 3.1.27. N′-Benzoyl-6-chloro-3-oxo-3,4-dihydroquinoxaline-2-carbohydrazide (**62**)

The title compound was prepared from **43** and benzoyl chloride using the general procedure II to afford **62** as a yellow solid (0.195 g, 78%), mp > 300 °C (dec). ^1^H-NMR δ (DMSO-*d*_6_): 7.39–7.46 (m, 2H), 7.53–7.65 (m, 3H), 7.82–7.94 (m, 3H), 10.90 (s, 1H, NH exch. with D_2_O), 11.01 (s, 1H, NH exch. with D_2_O), 12.98 (br s, 1H, NH exch. with D_2_O).

#### 3.1.28. N′-Benzoyl-6-trifluoromethyl-3-oxo-3,4-dihydroquinoxaline-2-carbohydrazide (**63**)

The title compound was prepared from **44** and benzoyl chloride using the general procedure II to afford **63** as a cream solid (0.244 g, 89%), mp = 321–325 °C. ^1^H-NMR δ (DMSO-*d*_6_): 7.51–7.72 (m, 5H), 7.95 (d, 2H, J = 7.6), 8.12 (d, 1H, J = 8.4), 10.92 (s, 1H, NH exch. with D_2_O), 11.03 (s, 1H, NH exch. with D_2_O), 12.93 (br s, 1H, NH exch.with D_2_O).

#### 3.1.29. N′-(4-(*tert*-Butyl)benzoyl-6-trifluoromethyl-3-oxo-3,4-dihydroquinoxaline-2-carbohydrazide (**64**)

The title compound was prepared from **44** and 4-*tert*-butylbenzoyl chloride using the general procedure II to afford **64** as a yellow solid (0.234 g, 74%), mp = 302–308 °C. ^1^H-NMR δ (DMSO-*d*_6_): 1.32 (s, 9H), 7.54 (d, 2H, J = 8.4 Hz), 7.67–7.71 (m, 2H), 7.87 (d, 2H, J = 8.4 Hz), 8.12 (d, 1H, J = 8.4 Hz), 10.84 (s, 1H, NH exch. with D_2_O), 10.99 (s, 1H, NH exch. with D_2_O), 13.09 (br s, 1H, NH exch. with D_2_O).

#### 3.1.30. N′-Cyclohexanecarbonyl-6-trifluoromethyl-3-oxo-3,4-dihydroquinoxaline-2-carbohydrazide (**65**)

The title compound was prepared from **44** and cyclohexanecarbonyl chloride using the general procedure II to afford **65** as a yellow solid (0.198 g, 71%), mp = 322–325 °C. ^1^H-NMR δ (DMSO-*d*_6_): 1.26–1.41 (m, 5H), 1.62–1.75 (m, 5H), 2.15–2.18 (m, 1 H), 7.65 (s, 1H), 7.69 (d, 1H, J = 8.8 Hz), 8.10 (d, 1H, J = 8.4 Hz), 10.48 (s, 1H, NH exch. with D_2_O), 11.05 (s, 1H, NH exch. with D_2_O), 13.10 (br s, 1H, NH exch. with D_2_O).

#### 3.1.31. N′-Benzoyl-3-oxo-7-(trifluoromethyl)-3,4-dihydroquinoxaline-2-carbohydrazide (**66**)

The title compound was prepared from **45** and benzoyl chloride using the general procedure II to afford **66** as a yellowish solid (0.242 g, 88%), mp > 300 °C (dec). ^1^H-NMR δ (DMSO-*d*_6_): 7.51–7.61 (m, 4H), 7.93–8.00 (m, 3H), 8.26 (s, 1H), 10.92 (s, 1H, NH exch. with D_2_O), 10.99 (s, 1H, NH exch. with D_2_O), 13.19 (br s, 1H, NH exch. with D_2_O).

#### 3.1.32. N′-(4-(*tert*-Butyl)benzoyl)-3-oxo-7-(trifluoromethyl)-3,4-dihydroquinoxaline-2-carbohydrazide (**67**)

The title compound was prepared from **45** and 4-*tert*-butylbenzoyl chloride using the general procedure II to afford **67** as a yellow solid (0.246 g, 78%), mp = 291–293 °C. ^1^H-NMR δ (DMSO-*d*_6_): 1.29 (s, 9H), 7.50–7.53 (m, 1H), 7.54 (d, 2H, J = 7.6 Hz), 7.89 (d, 2H, J = 7.6 Hz), 7.98 (d, 1H, J = 8.4 Hz), 8.25 (s, 1H), 10.85 (s, 1H, NH exch. with D_2_O), 10.98 (s, 1H, NH exch. with D_2_O), 13.19 (br s, 1H, NH exch. with D_2_O).

#### 3.1.33. N′-(Cyclohexanecarbonyl)-3-oxo-7-(trifluoromethyl)-3,4-dihydroquinoxaline-2-carbohydrazide (**68**)

The title compound was prepared from **45** and cyclohexanecarbonyl chloride using the general procedure II to afford **68** as a yellowish solid (0.145 g, 52%), mp = 313–315 °C. ^1^H-NMR δ (DMSO-*d*_6_): 2.16–2.18 (m, 5H), 1.16–1.27 (m, 5H), 2.33–2.35 (m, 1H), 7.54 (d, 1H, J = 8.8 Hz), 7.97 (d, 1H, J = 8.4 Hz), 8.23 (s, 1H), 10.50 (s, 1H, NH exch. with D_2_O), 11.23 (s, 1H, NH exch. with D_2_O), 13.19 (br s, 1H, NH exch. with D_2_O).

#### 3.1.34. (*E*)-N′-(2-(3-Oxo-3,4-dihydroquinoxalin-2(*1H*)-ylidene)acetyl)benzohydrazide (**71**)

The title compound was prepared from **70** and benzoyl chloride using the general procedure II to afford **71** as a yellow solid (0.087 g, 54%), mp = 265–268 °C. ^1^H-NMR δ (DMSO-*d*_6_): 5.78 (s, 1H), 6.96 (d, 1H, J = 7.2 Hz), 7.03 (t, 2H, J = 7.6 Hz), 7.19 (d, 1H, J = 7.6 Hz), 7.51 (t, 2H, J = 7.6 Hz), 7.58 (t, 1H, J = 7.6 Hz), 7.89 (d, 2H, J = 7.6 Hz), 9.10 (br s, 1H, NH exch. with D_2_O), 10.31 (br s, 1H, NH exch. with D_2_O), 11.55 (br s, 1H, NH exch. with D_2_O), 11.61 (br s, 1H, NH exch. with D_2_O).

#### 3.1.35. (*E)*-4-Chloro-N′-(2-(3-Oxo-3,4-dihydroquinoxalin-2(*1H*)-ylidene)acetyl)benzohydrazide (**72**)

The title compound was prepared from **70** and 4-chlorobenzoyl chloride using the general procedure II to afford **72** as a yellow solid (0.130 g, 73%), mp = 255–259 °C. ^1^H-NMR δ (DMSO-*d*_6_): 5.78 (s, 1H), 6.93–7.05 (m, 2H), 7.20 (d, 1H, J = 7.6 Hz), 7.32 (d, 1H, J = 8.0 Hz), 7.55–7.60 (m, 2H), 7.88–7.91 (m, 2H), 9.97 (br s, 1H, NH exch. with D_2_O), 10.41 (br s, 1H, NH exch. with D_2_O), 11.61 (br s, 2H, NH × 2 exch. with D_2_O).

#### 3.1.36. General Procedure III: Synthesis of Oxadiazole-Quinoxalines **4**–**28**

A solution of acylhydrazide **46**–**68, 71,** and **72** (0.50 mmol, 1 eq) in POCl_3_ (2.80 mL, 30 mmol, 60 eq) was stirred at 110 °C for 2 h under inert atmosphere (N_2_) and then poured into ice. The resulting precipitate was filtered and air-dried to yield desired compounds **4***–***28**.

#### 3.1.37. 2-(3-Chloroquinoxaline-2-yl)-5-phenyl-1,3,4-oxadiazole (**4**)

The title compound was prepared from **46** using the general procedure III to afford a yellowish solid (0.105 g, 68%), mp = 197.5–200 °C. ^1^H-NMR δ (DMSO-*d*_6_): 7.65–7.75 (m, 4H), 8.02–8.10 (m, 2H), 8.16–8.19 (m, 2H), 8.31–8.33 (m, 1H); ^13^C-NMR ppm (DMSO-*d*_6_): 123.22 (C), 127.53 (CH × 2), 128.50 (CH), 129.82 (CH), 130.13 (CH × 2), 132.42 (CH), 133.18 (CH), 134.12 (CH), 137.65 (C), 139.95 (C), 141.57 (C), 144.59 (C), 161.39 (C), 165.55 (C). MS (ESI): C_16_H_9_ClN_4_O requires *m*/*z* 308.05, found 309.05 [M+H]^+^. Anal. calcd for C_16_H_9_ClN_4_O: C, 62.25; H, 2.94; N, 18.15. Found: C, 62.07; H, 2.92; N, 17.98.

#### 3.1.38. 2-(3-Chloroquinoxaline-2-yl)-5-(p-tolyl)-1,3,4-oxadiazole (**5**)

The title compound was prepared from **47** using the general procedure III to afford a beige solid (0.131 g, 81%), mp = 185–190 °C. ^1^H-NMR δ (DMSO-*d*_6_): 2.45 (s, 3H), 7.51 (d, 2H, J_o_ = 8.0 Hz), 8.03–8.10 (m, 4H), 8.18 (d, 1H, J_o_ = 7.6 Hz), 8.33 (d, 1H, J_o_ = 7.6 Hz). ^13^C-NMR ppm (DMSO-*d*_6_): 21.20 (CH_3_), 119.98 (C), 127.01 (CH × 2), 128.02 (CH), 129.34 (CH), 130.20 (CH × 2), 131.92 (CH), 133.59 (CH), 137.22 (C), 139.47 (C), 141.06 (C), 143.03 (C), 144.11 (C), 160.70 (C), 165.16 (C). MS (ESI): C_17_H_11_ClN_4_O requires *m*/*z* 322.06, found 323.06 [M+H]^+^. Anal. calcd for C_17_H_11_ClN_4_O: C, 63.26; H, 3.44; N, 17.36. Found: C, 62.91; H, 3.42; N, 16.92.

#### 3.1.39. 2-(3-Chloroquinoxaline-2-yl)-5-(4-methoxyphenyl)-1,3,4-oxadiazole (**6**)

The title compound was prepared from **48** using the general procedure III to afford a beige solid (0.130 g, 77%), mp = 180–185 °C. ^1^H-NMR δ (DMSO-*d*_6_): 7.65–7.75 (m, 4H), 8.02–8.10 (m, 2H), 8.16–8.19 (m, 2H), 8.31–8.33 (m, 1H); ^13^C-NMR ppm (DMSO-*d*_6_): 3.89 (s, 3H), 7.23 (d, 2H, J_o_ = 8.8 Hz), 8.02–8.08 (m, 2H), 8.11 (d, 2H, J_o_ = 8.8 Hz), 8.17 (d, 1H, J_o_ = 7.6 Hz), 8.32 (d, 1H, J_o_ = 8.0 Hz). ^13^C-NMR ppm (DMSO-*d*_6_): 55.61 (OCH_3_), 114.96 (C), 115.05 (CH), 115.12 (CH), 128.00 (CH), 128.84 (CH), 128.98 (CH), 129.31 (CH), 131.89 (CH), 133.53 (CH), 137.22 (C), 139.46 (C), 141.02 (C), 144.07 (C), 160.45 (C), 162.60 (C), 165.03 (C). MS (ESI): C_17_H_11_ClN_4_O_2_ requires *m*/*z* 338.06, found 339.06 [M+H]^+^. Anal. calcd for C_17_H_11_ClN_4_O_2_: C, 60.28; H, 3.27; N, 16.54. Found: C, 59.45; H, 3.19; N, 16.32.

#### 3.1.40. 2-(3-Chloroquinoxaline-2-yl)-5-(4-chlorophenyl)-1,3,4-oxadiazole (**7**)

The title compound was prepared from **49** using the general procedure III to afford a yellowish solid (0.154 g, 90%), mp = 225–229 °C. ^1^H-NMR δ (DMSO-*d*_6_): 7.77 (d, 2H, J = 8.0 Hz), 8.05–8.10 (m, 2H), 8.19 (d, 3H, J = 7.6 Hz), 8.43–8.46 (m, 1H); ^13^C-NMR ppm (DMSO-*d*_6_): 121.66 (C), 128.03 (CH), 128.88 (CH × 2), 129.34 (CH), 129.86 (CH × 2), 131.99 (CH), 133.71 (CH), 137.07 (C), 137.46 (C), 139.46 (C), 141.11 (C), 144.11 (C), 161.04 (C), 164.34 (C). MS (ESI): C_16_H_8_Cl_2_N_4_O requires *m*/*z* 342.01, found 343.01 [M+H]^+^. Anal. calcd for C_16_H_8_Cl_2_N_4_O: C, 56.00; H, 2.35; N, 16.33. Found: C, 55.37; H, 2.19; N, 16.18.

#### 3.1.41. 2-(3-Chloroquinoxaline-2-yl)-5-(4-(*tert*-butyl)-phenyl)-1,3,4-oxadiazole (**8**)

The title compound was prepared from **50** using the general procedure III to afford a gray solid (0.166 g, 91%), mp = 173–175 °C. ^1^H-NMR δ (DMSO-*d*_6_): 7.77 (d, 2H, J = 8.0 Hz), 8.05–8.10 (m, 2H), 8.19 (d, 3H, J = 7.6 Hz), 8.43–8.46 (m, 1H); ^13^C-NMR ppm (DMSO-*d*_6_): 1.35 (s, 9H), 7.70 (d, 2H, J_o_ = 8.4 Hz), 8.00–8.07 (m, 2H), 8.08 (d, 2H, J_o_ = 8.4 Hz), 8.17 (dd, 1H, J_m_ = 1.2 Hz, J_o_ = 8.8 Hz), 8.31 (dd, 1H, J_m_ = 1.2 Hz, J_o_ = 8.8 Hz). ^13^C-NMR ppm (DMSO-d6): 30.78 (CH_3 ×_ 3), 34.93 (C), 120.03 (C), 126.23 (CH × 2), 126.50 (CH × 2), 128.02 (CH), 129.35 (CH), 131.92 (CH), 133.60 (CH), 137.23 (C), 139.47 (C), 141.08 (C), 144.12 (C), 155.74 (C), 160.73 (C), 165.09 (C). MS (ESI): C_20_H_17_ClN_4_O requires *m*/*z* 364.11, found 365.11 [M+H]^+^. Anal. calcd for C_20_H_17_ClN_4_O: C, 65.84; H, 4.70; N, 15.36. Found: C, 65.68; H, 4.58; N, 15.19.

#### 3.1.42. 2-(3-Chloroquinoxaline-2-yl)-5-cyclohexyl-1,3,4-oxadiazole (**9**)

The title compound was prepared from **51** using the general procedure III to afford a beige solid (0.152 g, 78%), mp = 198–100 °C. ^1^H-NMR δ (DMSO-*d*_6_): 1.30–1.36 (m, 1H), 1.40–1.50 (m, 2H), 1.61–1.70 (m, 3H), 1.78–1.81 (m, 2H), 2.10–2.13 (m, 2H), 3.14–3.20 (m, 1H), 7.99–8.08 (m, 2H), 8.15 (d, 1H, J_o_ = 8.4 Hz), 8.27 (d, 1H, J_o_ = 8.0 Hz). ^13^C-NMR ppm (DMSO-*d*_6_): 25.09 (CH_2_ × 2), 25.56 (CH_2_), 30.08 (CH_2_ × 2), 34.62 (CH), 128.46 (CH), 129.76 (CH), 132.32 (CH), 133.98 (CH), 137.85 (C), 139.89 (C), 141.53 (C), 144.54 (C), 160.98 (C), 171.38 (C). MS (ESI): C_16_H_15_ClN_4_O requires *m*/*z* 314.09, found 315.09 [M+H]^+^. Anal. calcd for C_16_H_15_ClN_4_O: C, 61.05; H, 4.80; N, 17.80. Found: C, 60.35; H, 4.61; N, 17.39.

#### 3.1.43. 2-(3-Chloroquinoxaline-2-yl)-5-(m-tolyl)-1,3,4-oxadiazole (**10**)

The title compound was prepared from **52** using the general procedure III to afford a gray solid (0.145 g, 90%), mp = 163–165 °C. ^1^H-NMR δ (DMSO-*d*_6_): 2.46 (s, 3H), 7.51–7.58 (m, 2H), 7.95 (d, 2H, J_o_ = 8.0 Hz), 8.02–8.09 (m, 2H), 8.17 (d, 1H, J_o_ = 8.0 Hz), 8.32 (d, 1H, J_o_ = 8.0 Hz). ^13^C-NMR ppm (DMSO-*d*_6_): 20.81 (CH_3_), 122.64 (C), 124.24 (CH), 127.22 (CH), 128.00 (CH), 129.33 (CH), 129.53 (CH), 131.90 (CH), 133.35 (CH), 133.60 (CH), 137.12 (C), 139.18 (C), 139.44 (C), 141.05 (C), 144.08 (C), 160.83 (C), 165.12 (C). MS (ESI): C_17_H_11_ClN_4_O requires *m*/*z* 322.06, found 323.06 [M+H]^+^. Anal. calcd for C_17_H_11_ClN_4_O: C, 63.26; H, 3.44; N, 17.36. Found: C, 62.87; H, 3.41; N, 16.90.

#### 3.1.44. 2-(3-Chloroquinoxaline-2-yl)-5-(3-methoxyphenyl)-1,3,4-oxadiazole (**11**)

The title compound was prepared from **53** using the general procedure III to afford a gray solid (0.157 g, 93%), mp = 144–148 °C. ^1^H-NMR δ (DMSO-*d*_6_): 7.65–7.75 (m, 4H), 8.02–8.10 (m, 2H), 8.16–8.19 (m, 2H), 8.31–8.33 (m, 1H); ^13^C-NMR ppm (DMSO-*d*_6_): 3.90 (s, 3H), 7.29 (dd, 1H, J_m_ = 2.0 Hz, J_o_ = 7.6 Hz), 7.59–7.63 (m, 2H), 7.73 (d, 1H, J_o_ = 7.6 Hz) 8.02–8.10 (m, 2H), 8.17 (d, 1H, J_o_ = 7.6 Hz), 8.36 (d, 1H, J_o_ = 8.0 Hz). ^13^C-NMR ppm (DMSO-*d*_6_): 55.51 (OCH_3_), 111.76 (CH), 118.63 (CH), 119.35 (CH), 123.88 (C), 128.01 (CH), 129.36 (CH), 131.01 (CH), 131.92 (CH), 133.65 (CH), 137.13 (C), 139.46 (C), 141.10 (C), 144.10 (C), 159.74 (C), 160.91 (C), 164.91 (C). MS (ESI): C_17_H_11_ClN_4_O_2_ requires *m*/*z* 338.06, found 339.06 [M+H]^+^. Anal. calcd for C_17_H_11_ClN_4_O_2_: C, 60.28; H, 3.27; N, 16.54. Found: C, 59.50; H, 3.20; N, 16.57.

#### 3.1.45. 2-(3-Chloroquinoxaline-2-yl)-5-(trimethoxyphenyl)-1,3,4-oxadiazole (**12**)

The title compound was prepared from **54** using the general procedure III to afford a green solid (0.136 g, 68%), mp = 204–208 °C. ^1^H-NMR δ (DMSO-*d*_6_): 7.65–7.75 (m, 4H), 8.02–8.10 (m, 2H), 8.16–8.19 (m, 2H), 8.31–8.33 (m, 1H); ^13^C-NMR ppm (DMSO-*d*_6_): 3.80 (s, 3H), 3.94 (s, 6H), 7.40 (s, 2H), 8.04 (t, 1H, J_o_ = 8.8 Hz), 8.05 (t, 1H, J_o_ = 8.0 Hz), 8.18 (d, 1H, J_o_ = 8.4 Hz), 8.33 (d, 1H, J_o_ = 8.0 Hz). ^13^C-NMR ppm (DMSO-*d*_6_): 56.24 (OCH_3_), 56.33 (OCH_3_), 60.29 (OCH_3_), 104.43 (CH × 2), 117.82 (C), 128.03 (CH), 129.36 (CH), 131.91 (CH), 133.64 (CH), 137.19 (C), 139.44 (C), 141.11 (C), 141.20 (C), 144.14 (C), 153.57 (C × 2), 160.76 (C), 165.00 (C). MS (ESI): C_19_H_15_ClN_4_O_4_ requires *m*/*z* 398.08, found 399.08 [M+H]^+^. Anal. calcd for C_19_H_15_ClN_4_O_4_: C, 57.22; H, 3.79; N, 14.05. Found: C, 58.13; H, 3.81; N, 14.75.

#### 3.1.46. 2-(3-Chloroquinoxaline-2-yl)-5-(2-chlorophenyl)-1,3,4-oxadiazole (**13**)

The title compound was prepared from **55** using the general procedure III to afford a yellowish solid (0.154 g, 59%), mp = 190–192 °C. ^1^H-NMR δ (DMSO-*d*_6_): 7.66 (t, 1H, J = 7.2 Hz), 7.74 (t, 1H, J = 8.0 Hz), 7.81 (d, 1H, J = 8.0 Hz), 8.03–8.15 (m, 3H), 8.19 (d, 1H, J = 8.0 Hz), 8.32 (d, 1H, J = 8.0 Hz). ^13^C-NMR ppm (DMSO-*d*_6_): 122.04 (C), 128.03 (CH), 128.06 (CH), 129.41 (CH), 131.26 (CH), 131.68 (CH), 131.97 (CH), 132.16 (C), 133.76 (CH), 133.80 (CH), 137.08 (C), 139.50 (C), 141.21 (C), 144.13 (C), 161.24 (C), 163.23 (C). MS (ESI): C_16_H_8_Cl_2_N_4_O requires *m*/*z* 342.01, found 343.01 [M+H]^+^. Anal. calcd for C_16_H_8_Cl_2_N_4_O: C, 56.00; H, 2.35; N, 16.33. Found: C, 55.43; H, 2.21; N, 16.38.

#### 3.1.47. 2-(3-Chloroquinoxaline-2-yl)-5-(3-chlorophenyl)-1,3,4-oxadiazole (**14**)

The title compound was prepared from **56** using the general procedure III to afford a yellowish solid (0.069 g, 40%), mp = 190–193 °C. ^1^H-NMR δ (DMSO-*d*_6_): 7.74 (t, 1H, J = 8.0 Hz), 7.82 (d, 1H, J = 8.0 Hz), 8.04–8.10 (m, 2H), 8.13–8.15 (m, 2H), 8.20 (d, 1H, J = 8.0 Hz), 8.35 (d, 1H, J = 8.0 Hz). ^13^C-NMR ppm (DMSO-*d*_6_): 124.72 (C), 125.81 (CH), 126.44 (CH), 128.04 (CH), 129.39 (CH), 131.74 (CH), 131.99 (CH), 132.51 (CH), 133.75 (CH), 134.21 (C), 137.02 (C), 139.47 (C), 141.14 (C), 144.11 (C), 161.17 (C), 163.94 (C). MS (ESI): C_16_H_8_Cl_2_N_4_O requires *m*/*z* 342.01, found 343.01 [M+H]^+^. Anal. calcd for C_16_H_8_Cl_2_N_4_O: C, 56.00; H, 2.35; N, 16.33. Found: C, 55.37; H, 2.19; N, 16.34.

#### 3.1.48. 2-(3-Chloroquinoxaline-2-yl)-5-(2,4-dichlorophenyl)-1,3,4-oxadiazole (**15**)

The title compound was prepared from **57** using the general procedure III to afford a yellowish solid (0.179 g, 95%), mp = 247–251 °C. ^1^H-NMR δ (DMSO-*d*_6_): 7.76 (d, 1H, J = 8.4 Hz), 8.02–8.12 (m, 3H), 8.15–8.21 (m, 2H), 8.32 (d, 1H, J = 7.2 Hz). ^13^C-NMR ppm (DMSO-*d*_6_): 120.98 (C), 126.40 (CH), 127.14 (CH), 127.17 (CH), 127.85 (CH), 131.20 (CH), 131.47 (CH), 132.35 (CH), 134.52 (C), 134.89 (C), 139.08 (C), 139.27 (C), 141.07 (C), 146.40 (C), 161.15 (C), 163.98 (C). MS (ESI): C_16_H_7_Cl_3_N_4_O requires *m*/*z* 342.01, found 375.97 [M+H]^+^. Anal. calcd for C_16_H_7_Cl_3_N_4_O: C, 50.89; H, 1.87; N, 14.84. Found: C, 50.16; H, 1.85; N, 14.56.

#### 3.1.49. 2-(3-Chloro-6-methoxyquinoxaline-2-il)-5-phenyl-1,3,4-oxadiazole (**16**)

The title compound was prepared from **58** using the general procedure III to afford a gray solid (0.168 g, 99%), mp = 198–200 °C. ^1^H-NMR δ (DMSO-*d*_6_): 4.03 (s, 3H), 7.57 (d, 1H, J_m_ = 2.4 Hz), 7.66–7.72 (m, 5H), 8.16 (dd, 2H, J_m_ = 1.2 Hz, J_o_ = 6.4 Hz), 8.22 (d, 1H, J_o_ = 9.2 Hz). ^13^C-NMR ppm (DMSO-*d*_6_): 56.48 (OCH_3_), 106.06 (CH), 122.83 (C), 125.01 (CH), 126.97 (CH × 2), 129.63 (CH × 2), 130.55 (CH), 132.59 (CH). 133.90 (C), 135.77 (C), 143.46 (C), 144.46 (C), 161.04 (C), 163.20 (C), 164.86 (C). MS (ESI): C_17_H_11_ClN_4_O_2_ requires *m*/*z* 338.06, found 339.06 [M+H]^+^. Anal. calcd for C_17_H_11_ClN_4_O_2_: C, 60.28; H, 3.27; N, 16.54. Found: C, 58.96; H, 3.19; N, 16.16.

#### 3.1.50. 2-(3-Chloro-7-methoxyquinoxaline-2-il)-5-phenyl-1,3,4-oxadiazole (**17**)

The title compound was prepared from **59** using the general procedure III to afford a gray solid (0.117 g, 69%), mp = 186–190 °C. ^1^H-NMR δ (DMSO-*d*_6_): 4.02 (s, 2H), 7.65–7.67 (m, 1H), 7.68–7.70 (m, 2H), 7.71–7.72 (m, 2H), 8.06 (d, 1H, J = 9.2 Hz), 8.16 (dd, 2H, J_m_ = 1.2 Hz, J_o_ = 8.0 Hz).^13^C-NMR ppm (DMSO-*d*_6_): 56.33 (OCH_3_), 106.86 (CH), 122.77 (CH), 126.69 (CH), 126.91 (CH), 127.00 (CH), 129.03 (CH), 129.64 (CH × 2), 132.66 (CH), 136.79 (C), 137.34 (C), 141.36 (C), 141.39 (C), 161.01 (C), 161.54 (C), 165.00 (C). MS (ESI): C_17_H_11_ClN_4_O_2_ requires *m*/*z* 338.06, found 339.06 [M+H]^+^. Anal. calcd for C_17_H_11_ClN_4_O_2_: C, 60.28; H, 3.27; N, 16.54. Found: C, 59.24; H, 3.23; N, 16.20.

#### 3.1.51. 2-(3-Chloro-7-methoxyquinoxaline-2-yl)-(4-(*tert*-butyl)phenyl)-5-1,3,4-oxadiazole (**18**)

The title compound was prepared from **60** using the general procedure III to afford a white solid (0.079 g, 69%), mp = 181–182 °C. ^1^H-NMR δ (DMSO-*d*_6_): 1.32 (s, 9H), 3.95 (s, 3H), 7.48–7.49 (m, 2H), 7.51 (d, 2H, J = 8.0 Hz), 7.92 (d, 1H, J = 9.2 Hz), 8.10 (d, 2H, J = 8.4 Hz). ^13^C ppm (DMSO-*d*_6_): 31.12 (CH_3_ × 3), 35.19 (C), 56.11 (OCH_3_), 106.68 (CH), 120.52 (C), 126.18 (CH × 2), 126.91 (CH), 127.37 (CH × 2), 129.23 (CH), 137.37 (C), 138.30 (C), 142.08 (C), 156.12 (C), 161.10 (C), 161.83 (C), 166.04 (C). MS (ESI): C_21_H_19_ClN_4_O_2_ requires *m*/*z* 394.12, found 395.12 [M+H]^+^. Anal. calcd for C_21_H_19_ClN_4_O_2_: C, 63.88; H, 4.85; N, 14.19. Found: C, 63.13; H, 4.73; N, 14.02.

#### 3.1.52. 2-(3-Chloro-7-metoxyquinoxaline-2-il)-5-cyclohexyl-1,3,4-oxadiazole (**19**)

The title compound was prepared from **61** using the general procedure III to afford a white solid (0.031 g, 18%), mp = 133–134 °C. ^1^H-NMR δ (DMSO-*d*_6_): 1.43–1.53 (m, 4H), 1.74–1.84 (m, 2H), 1.90–1.95 (m, 2H), 2.22–2.25 (m, 2H), 3.10–3.16 (m, 1H), 4.02 (s, 3H), 7.53 (d, 1H, J_m_ = 2.8 Hz), 7.57 (dd, 1H, J_m_ = 2.4 Hz, J_o_ = 9.2 Hz), 7.99 (d, 1H, J = 9.2 Hz). ^13^C ppm (DMSO-*d*_6_): 25.38 (CH_2_ × 2), 25.54 (CH_2_), 30.17 (CH_2_ × 2), 35.36 (CH), 56.10 (CH_3_), 106.65 (CH), 126.83 (CH), 129.21 (CH), 137.54 (C), 138.26 (C), 141.99 (C), 142.30 (C), 161.15 (C), 161.79 (C), 171.65 (C). MS (ESI): C_17_H_17_ClN_4_O_2_ requires *m*/*z* 334.10, found 335.10 [M+H]^+^. Anal. calcd for C_17_H_17_ClN_4_O_2_: C, 59.22; H, 4.97; N, 16.25. Found: C, 59.07; H, 4.96; N, 16.20.

#### 3.1.53. 2-(3,6-Dichloroquinoxaline-2-il)-5-phenyl-1,3,4-oxadiazole (**20**)

The title compound was prepared from **62** using the general procedure III to afford a pink solid (0.125 g, 73%), mp = 227–228 °C. ^1^H-NMR δ (DMSO-*d*_6_): 7.68–7.73 (m, 3H), 8.06–8.08 (d, 1H, J = 8.8 Hz), 8.16–8.18 (d, 2H, J = 6.8 Hz), 8.34–8.36 (d, 2H, J = 9.6 Hz). ^13^C-NMR ppm (DMSO-*d*_6_): 122.69 (C), 126.94 (CH), 127.07 (CH × 2), 129.67 (CH × 2), 131.11 (CH), 132.54 (CH), 132.75 (CH), 137.47 (C), 138.04 (C), 138.26 (C), 141.42 (C), 145.32 (C), 160.81 (C), 165.11 (C). MS (ESI): C_16_H_8_Cl_2_N_4_O requires *m*/*z* 342.01, found 343.01 [M+H]^+^. Anal. calcd for C_16_H_8_Cl_2_N_4_O: C, 56.00; H, 2.32; N, 16.33. Found: C, 55.87; H, 2.31; N, 16.28.

#### 3.1.54. 2-(3-Chloro-6-(trifluoromethyl)quinoxaline-2-yl)-5-phenyl-1,3,4-oxadiazole (**21**)

The title compound was prepared from **63** using the general procedure III to afford a gray solid (0.75 g, 40%), mp = 255–258 °C. ^1^H-NMR δ (CDCl_3_-*d*_6_): 7.76–7.69 (m, 3H), 8.24 (d, 1H, J = 6.4 Hz), 8.30 (dd, 1H, J_m_ = 1.6 Hz, J_o_ = 8.8 Hz), 8.55 (d, 1H, J = 8.8 Hz), 8.63 (s, 1H). ^13^C ppm (DMSO-*d*_6_): 122.63 (C), 124.05 (C-CF_3_), 126.16 (CH), 127.03 (CH), 127.14 (CH × 2), 129.71 (CH × 2), 131.24 (CH), 132.85 (CH), 132.61 (C), 139.52 (C), 140.19 (C), 140.73 (C), 145.82 (C), 160.77 (C), 165.27 (C). MS (ESI): C_17_H_8_ClF_3_N_4_O requires *m*/*z* 376.03, found 377.03 [M+H]^+^. Anal. calcd for C_17_H_8_ClF_3_N_4_O: C, 54.20; H, 2.14; N, 14.87. Found: C, 54.36; H, 2.16; N, 14.91.

#### 3.1.55. 2-(3-Chloro-6-(trifluoromethyl)quinoxaline-2-yl)-5-(4-*tert*-butyl)phenyl)-1,3,4-oxadiazole (**22**)

The title compound was prepared from **64** using the general procedure III to afford a yellow solid (0.50 g, 23%), mp = 212–215 °C. ^1^H-NMR δ (CDCl_3_-*d*_6_): 1.37 (s, 9H), 7.53 (d, 2H, J_o_ = 8.8 Hz), 8.00 (dd, 1H, J_m_ = 1.6Hz, J_o_ = 8.8 Hz), 8.12 (d, 2H, J = 8.8 Hz), 8.37–8.39 (m, 2H). ^13^C ppm (DMSO-*d*_6_): 25.85 (CH_3_ × 3), 29.99 (C), 114.99 (C), 121.15 (CH), 121.03 (CH × 2), 121.79 (CH), 122.24 (CH × 2), 125.84 (CH), 129.13 (C), 130.88 (C), 132.94 (C), 134.32 (C), 135.63 (C), 135.91 (C), 151.24 (C), 155.26 (C), 161.25 (C). MS (ESI): C_21_H_16_ClF_3_N_4_O requires *m*/*z* 432.10, found 433.10 [M+H]^+^. Anal. calcd for C_21_H_16_ClF_3_N_4_O: C, 58.27; H, 3.73; N, 12.94. Found: C, 58.33; H, 3.74; N, 12.99.

#### 3.1.56. 2-(3-Chloro-6-(trifluoromethyl)quinoxaline-2-yl)-5-cyclohexyl-1,3,4-oxadiazole (**23**)

The title compound was prepared from **65** using the general procedure III to afford a white solid (0.50 g, 26%), mp = 161–163 °C. ^1^H-NMR δ (CDCl_3_-*d*_6_): 1.35–1.55 (m, 3H), 1.59 (s, 3H), 1.76–1.80 (m, 2H), 1.92–1.96 (m, 1H), 2.22–2.26 (m, 1H), 3.11–3.19 (m, 1H), 8.07 (dd, 1H, J*_m_* = 1.6 Hz, J*_o_* = 8.8 Hz), 8.41 (d, 1H, J = 8.8 Hz), 8.44 (s, 1H). ^13^C ppm (DMSO-*d*_6_): 25.33 (CH_2_ × 2), 25.48 (CH_2_), 30.13 (CH_2_ × 2), 35.36 (CH), 126.35 (CH), 126.98 (CH), 131.01 (CH), 134.50 (C), 139.75 (C), 140.84 (C), 140.94 (C), 146.00 (C), 160.56 (C), 171.93 (C), 173.00 (C). MS (ESI): C_17_H_14_ClF_3_N_4_O requires *m*/*z* 382.08, found 383.08 [M+H]^+^. Anal. calcd for C_17_H_14_ClF_3_N_4_O: C, 53.34; H, 3.69; N, 14.64. Found: C, 53.48; H, 3.71; N, 14.71.

#### 3.1.57. 2-(3-Chloro-7-(trifluoromethyl)quinoxaline-2-yl)-5-phenyl-1,3,4-oxadiazole (**24**)

The title compound was prepared from **66** using the general procedure III to afford a yellowish solid (0.71 g, 38%), mp = 150–152 °C. ^1^H-NMR δ (DMSO-*d*_6_): 7.66–7.75 (m, 3H), 8.17 (dd, 2H, J*_o_* = 8.4 Hz, J*_m_* = 1.6 Hz), 8.33 (dd, 1H, J*_o_* = 8.8 Hz, J*_m_* = 2 Hz), 8.39 (d, 1H, J = 8.8 Hz). ^13^C ppm (DMSO-*d*_6_): 123.34 (C-CF_3_), 127.05 (CH × 2), 127.41 (CH), 128.55 (CH), 129.65 (CH × 2), 129.96 (CH), 130.80 (CH), 131.61 (CF_3_, q, J = 132 Hz), 132.81 (CH), 138.54 (C), 138.83 (C), 142.26 (C), 146.56 (C), 160.66 (C), 165.19 (C). MS (ESI): C_17_H_8_ClF_3_N_4_O requires *m*/*z* 376.03, found 377.03 [M+H]^+^. Anal. calcd for C_17_H_8_ClF_3_N_4_O: C, 54.20; H, 2.14; N, 14.87. Found: C, 54.32; H, 2.15; N, 14.90.

#### 3.1.58. 2-(3-Chloro-7-(trifluoromethyl)quinoxaline-2-yl)-5-(4-*tert*-butyl)phenyl)-1,3,4-oxadiazole (**25**)

The title compound was prepared from **67** using the general procedure III to afford a yellow solid (0.43 g, 20%), mp = 198–200 °C. ^1^H-NMR δ (DMSO-*d*_6_): 1.36 (1, 9H), 7.73 (d, 2H, J = 7.6 Hz), 8.11 (d, 2H, J = 7.6 Hz), 8.39 (d, 1H, J = 8.8 Hz), 8.76 (s, 1 H). ^13^C ppm (DMSO-*d*_6_): 30.77 (CH_3_ × 3), 34.95 (C), 119.91 (C), 123.61 (C-CF_3_), 126.54 (CH × 2), 126.97 (CH × 2), 127.42 (CH), 128.51 (CH), 129.98 (CH), 131.38 (CF_3_, q, J = 128 Hz), 138.58 (C), 138.94 (C), 142.29 (C), 146.62 (C), 155.92 (C), 160.52 (C), 165.25 (C). MS (ESI): C_21_H_16_ClF_3_N_4_O requires *m*/*z* 432.10, found 433.10 [M+H]^+^. Anal. calcd for C_21_H_16_ClF_3_N_4_O: C, 58.27; H, 3.73; N, 12.94. Found: C, 58.35; H, 3.74; N, 12.98.

#### 3.1.59. 2-(3-Chloro-7-(trifluoromethyl)quinoxaline-2-yl)-5-cyclohexyl-1,3,4-oxadiazole (**26**)

The title compound was prepared from **68** using the general procedure III to afford a white solid (0.90 g, 47%), mp = 164–166 °C. ^1^H-NMR δ (DMSO-*d*_6_): 1.34–1.42 (m, 1H), 1.48–1.65 (m, 2H), 1.68–1.79 (m, 3H), 1.80–1.99 (m, 2H), 2.11–2.14 (m, 2H), 3.19 (s, 1H), 8.31(d, 1 H, J = 8.8 Hz), 8.37(d, 1H, J = 8.8 Hz), 8.71 (s, 1H). ^13^C ppm (DMSO-*d*_6_): 24.59 (CH_2_ × 2), 25.08 (CH_2_), 29.58 (CH_2_ × 2), 34.15 (CH), 123.35 (C-CF_3_), 127.35 (CH), 128.44 (CH), 129.95 (CH), 131.22 (CF_3_, q, J = 128 Hz,), 138.54 (C), 139.09 (C), 142.28 (C), 146.57 (C), 160.32 (C), 171.10 (C). MS (ESI): C_17_H_14_ClF_3_N_4_O requires *m*/*z* 382.08, found 383.08 [M+H]^+^. Anal. calcd for C_17_H_14_ClF_3_N_4_O: C, 53.34; H, 3.69; N, 14.64. Found: C, 53.45; H, 3.70; N, 14.70.

#### 3.1.60. 2-((3-Chloroquinoxaline-2-yl)methyl)-5-phenyl-1,3,4-oxadiazole (**27**)

The title compound was prepared from **71** using the general procedure III to afford an orange solid (0.073 g, 45%) after flash chromatography (hexane/EtOAc 6/4) purification, mp = 161–165 °C. ^1^H-NMR δ (DMSO-*d*_6_): 4.94 (s, 2H), 7.60–7.64 (m, 3H), 7.89–7.95 (m, 2H), 7.99 (d, 2H, J = 8.0 Hz), 8.07 (t, 2H, J = 8.0 Hz). ^13^C-NMR ppm (DMSO-*d*_6_): 32.45 (CH_2_), 123.25 (C), 126.47 (CH × 2), 127.84 (CH), 128.53 (CH), 129.46 (CH × 2), 131.03 (CH), 131.44 (CH), 132,01 (CH), 139.94 (C), 140.61 (C), 146.53 (C), 149.33 (C), 162.99 (C), 164.55 (C). MS (ESI): C_17_H_11_ClN_4_O requires *m*/*z* 322.06, found 323.06 [M+H]^+^. Anal. calcd for C_17_H_11_ClN_4_O: C, 63.26; H, 3.44; N, 17.36. Found: C, 62.92; H, 3.41; N, 17.22.

#### 3.1.61. 2-((3-Chloroquinoxaline-2-yl)methyl)-5-(4-chlorophenyl)-1,3,4-oxadiazole (**28**)

The title compound was prepared from **72** using the general procedure III to afford an orange solid (0.046 g, 26%) after flash chromatography (hexane/EtOAc 6/4) purification, mp = 189–193 °C. ^1^H-NMR δ (DMSO-*d*_6_): 4.94 (s, 2H), 7.67 (d, 2H, J = 8.8 Hz), 7.88–7.96 (m, 2H), 7.99 (d, 2H, J = 8.8 Hz), 8.05–8.10 (m, 2H). ^13^C-NMR ppm (DMSO-*d*_6_): 32.92 (CH_2_), 122.59 (C), 128.32 (CH), 128.78 (CH × 2), 129.01 (CH), 130.07 (CH), 130.13 (CH), 131.51 (CH), 131.93 (CH), 137.21 (C), 140.41 (C), 141.10 (C), 147.00 (C), 149.73 (C), 163.67 (C), 164.30 (C). MS (ESI): C_17_H_10_Cl_2_N_4_O requires *m/z* 356.02, found 357.02 [M+H]^+^. Anal. calcd for C_17_H_10_Cl_2_N_4_O: C, 57.16; H, 2.82; N, 15.69. Found: C, 56.88; H, 2.81; N, 15.

### 3.2. Biological Methods

#### 3.2.1. Cell Cultures

Human leukemic tumor cells (NB4, MOLT-4, JURKAT, and HL60) were maintained in RPMI 1640 medium supplemented with 10% FBS, L-glutamine (2 mM), penicillin (100 U/mL), and streptomycin (100 μg/mL). Human neuroblastoma cells (SH-SY5Y) were cultured in DMEM F12 and EMEM medium supplemented with 10% FBS, L-glutamine (2 mM), penicillin (100 U/mL), and streptomycin (100 μg/mL). Hepatocellular carcinoma (MAHLAVU), human breast adenocarcinoma (MDA-MB-231), and human melanoma cells (A375) were cultured in DMEM–High Glucose medium supplemented with 10% FBS, penicillin (100 U/mL), and streptomycin (100 μg/mL). All cell lines were maintained in a humidified atmosphere (5% CO_2_) at 37 °C and were subcultured three times a week.

#### 3.2.2. Preparation of Compounds

All tested compounds were dissolved in DMSO to a final concentration of 10 mM and stored at −20 °C until use. Diluted solutions from 10 μM to 100 nM were performed in cell medium.

#### 3.2.3. MTS Assay

The MTS assay was performed to determine cells vitality according to the manufacturer’s protocol from the CellTiter 96 Aqueous One Solution Cell Proliferation Assay. All cell lines (NB4, JURKAT, SH-SY5Y, MAHLAVU, A375, and lymphocytes) were plated in 96-multiwell plates. 5-fluorouracil (5-FU) was used as a positive control, and all tested compounds were added for 24, 48, and 72 h and, at the end of the incubation period, 20 μL of MTS solution was added to each well. The optical density of each well was read on a spectrophotometer at 490 nm.

#### 3.2.4. Human Lymphocytes Isolation

Human lymphocytes were isolated from EDTA whole blood samples collected from volunteers of the AVIS Blood Bank of Ferrara after written informed consent according to the CE 240/2024/Sper/UniFe protocol. Briefly, blood was centrifuged at 2200 rpm for 15 min on a Ficoll–Hypaque density gradient. Platelet Rich Plasma (PRP) was removed, and Peripheral Blood Mononuclear Cells (PBMCs) were isolated and removed from the Ficoll–Hypaque gradient. Subsequently, cells were washed twice with PBS and finally transferred to a T25 culture flask containing RPMI 1640 + FBS + Pen/Strep medium and placed in a humidified incubator (5% CO_2_) for 24 h at 37 °C to remove adherent monocytes from the lymphocyte culture.

#### 3.2.5. Flow Cytometry Analysis

Cytotoxic effects and cell cycle alterations were assessed using flow cytometry in JURKAT, SH-SY5Y96, and human lymphocytes treated with 5-FU, compounds **24**, **25**, and **26**, or DMSO (10 µM). Cytotoxicity was evaluated by measuring the percentage of viable, apoptotic, and necrotic cells after 6 h of treatment, while the effects on the cell cycle were assessed after 24 h. At each time point, cells were collected, and adherent cells (SH-SY5Y) were detached using Trypsin-EDTA (0.5%), mixed with the previously collected culture medium, and then centrifuged at 1200 rpm for 8 min. Cytotoxicity was evaluated after 6 h of treatment using Annexin V-FITC/7-AAD kit (Annexin V-FITC/7-AAD Kit, Beckman Coulter, Brea, CA, USA), as indicated by the manufacturer’s protocol. Briefly, 1 × 10^6^ cells were washed with 1X PBS and centrifuged for 5 min at 500× *g* at 4 °C. After discarding the supernatant, cells were resuspended in ice-cold 1X-Binding Buffer (200 μL) and incubated on ice with 10 μL of Annexin A5-FITC solution and 20 μL of 7-AAD Viability Die for 15 min in the dark. After the incubation, 400 μL of ice-cold 1X-Binding Buffer was added to the cells, and samples were analyzed using flow cytometry. Live cells were double negative for Annexin-V and 7-AAD (AnnexinV^−^/7-AAD^−^), apoptotic cells were positive for Annexin V and negative for 7-AAD (AnnexinV^+^/7-AAD^−^), and necrotic cells were double positive for Annexin V and 7-AAD (AnnexinV^+^/7-AAD^+^).

For the cell cycle analysis, cells were incubated with 70% ethanol for 1 h at 4 °C. The cells were washed with PBS 1X and incubated with PBS containing 20 μg/mL propidium iodide (PI) and 100 μg/mL RNAse-A for 15 min at room temperature in the dark before analysis. All the analyses were performed by using a CytoFLEX flow cytometer (Beckman Coulter, Brea, CA, USA) equipped with CytExpert software version 2.4 (Beckman Coulter).

#### 3.2.6. Statistical Analysis

All values in the figures and text are expressed as mean ± standard error (SEM) or standard deviation (SD), as indicated in figure legends of independent experiments. Data sets were examined by one-way analysis of variance (ANOVA) followed by Sidak’s or Tukey’s multiple comparisons tests. A *p*-value less than 0.05 was considered statistically significant.

## 4. Conclusions

The preliminary results in the full NCI 60 cell panel allowed some structure–activity relationship considerations. Firstly, we evaluated the effect of the phenyl ring on C-5 position of the 1,3,4-oxadiazole nucleus synthesizing derivative **4**. The compound showed good cytotoxic but limited to MCF7 and MDA-MB-468 cell lines. Therefore, we introduced in the phenyl ring electron-donating (**5**, **6**, and **12**) or electron-withdrawing (**7**, **14**, and **15**) groups obtaining compounds with poor-to-low activity. The introduction of a methyl group (**10**) or a methoxy group (**11**) in 3-position and that of a chlorine atom (**13**) in the 2-position cause a slight increase in the activity. On the other hand, the 4*-tert*-butylphenyl (**8**) and cyclohexyl (**9**) substituents on C-5 of oxadiazole proved better anticancer results. Then, we evaluated the effect of the introduction of a substituent in position C-6 or C-7 in the quinoxaline moiety of compounds **4**, **8**, and **9** due to their activity. The introduction of methoxy group in position C-6 (**16**) or C-7 (**17**) of the quinoxaline in the basic term has no effect, whereas the 2-(3-chloro-7-methoxy-quinoxaline-2-yl)-5-(4-*tert*-butylphenyl)-1,3,4-oxadiazole (**18**) and its -5-cyclohexyl (**19**) analog again exhibited moderate-to-good activity against various cell lines. A different trend was observed in the series of trifluoromethyl derivatives compared to the previous ones; thus, when the trifluoromethyl group is in C-6 position of quinoxaline (**21**–**23**), only the derivative with the cyclohexyl ring (**23**) on the oxadiazole elicited significant GI% values against leukemia, non-small cell lung, colon, and ovarian cancer. On the other hand, 2-(3-chloro-7-(trifluoromethyl)quinoxaline-2-yl)-5-phenyl-1,3,4-oxadiazole (**24**) and its 4-*tert*-butylphenyl (**25**) and cyclohexyl (**26**) analogs emerge as the most interesting compounds within this small library, resulting derivative **24** with the highest GI_50_ (1.85 μM, MCF7 cell line) in the whole series and with an MG_MID Log_10_ GI_50_ = −5.02, which accounts for the highest micromolar activity against all cell lines.

Other modifications of the quinoxaline-oxadiazole template, in the form of introduction of a chlorine in 6 position of the bicyclic system (**20**) and of a methylene bridge between the two heterocycles (**27**, **28**), were unfavorable for anticancer activity.

From this preliminary structure–activity relationship study, we can conclude that the main feature responsible for good anticancer activity in this newly synthesized series of 2-(3-chloro-quinoxaline-2-yl)-5-substituted-1,3,4-oxadiazoles is the presence of a 7-trifluoromethyl group in the quinoxaline ring resulting the nature of substituent in the C-5 position of 1,3,4-oxadiazole not as stringent.

In general, by evaluating the cytotoxicity by MTS assay on a selection of novel 25 compounds in other tumor cell lines not included in the full NCI 60 cell panel, we found that most of them showed a significant antiproliferative and cytotoxic effect when tested at a single dose of 10 µM with the most potent results in leukemic cells (NB4 and JURKAT) in comparison to solid tumor cells (SH-SY-5Y, A375, and MAHLAVU cells). Compounds **24**, **25**, and **26** are confirmed to be the most interesting for their activity across all the investigated cell lines. These compounds were also tested for potential cytotoxicity in human lymphocytes chosen as a non-tumorigenic cellular model. In these cells, the cytotoxic effects were less severe than those observed in cancer cell lines.

To further investigate the mechanisms underlying cytotoxicity, we assessed the ability of these compounds to induce apoptosis, necrosis, and cell cycle alterations. After 6 h of treatment, a significant increase in phosphatidylserine exposure was observed in cells treated with compounds **25** and **26,** compared to those treated with 5-FU. This suggests that the compounds exert a considerably stronger antitumor effect than 5-FU in both cell lines. No significant differences were found between cancer cell lines in adhesion and those in suspension. Moreover, after 24 h of treatment, both compounds induced significant DNA fragmentation, consistent with classical apoptosis induction. However, other cell death mechanisms, such as caspase-independent necrosis or senescence cannot be excluded, particularly for compound **26**.

We also evaluated the effects of our compounds on the cell cycle in cancer cell lines, which actively proliferate in vitro under optimal growth conditions. Treatment with compounds **25** (in JURKAT and SH-SY5Y cells) and **26** (in JURKAT cells only) led to an accumulation of cells in S-phase, comparable to the effects of 5-FU, which inhibits DNA synthesis by depleting thymidylate. This suggests that the compounds may not induce G1/S checkpoint crossing but rather slow down or arrest the progression of proliferating tumor cells in the S-phase. Considering PHA-activated lymphocytes, only compound **24,** and to a lesser extent compound **25**, affected the cell cycle, inducing a modest increase in the number of cells in the S and G2/M phases.

When comparing the effects of the compounds on JURKAT cells (tumorigenic T lymphoblasts) versus healthy lymphocytes, compound **26** is the only one that appears to have different effects on these two cell types. Flow cytometry data revealed that compound **26** had a higher efficiency in JURKAT cells, with a significant percentage of apoptotic cells observed after 6 h of treatment. Additionally, compound **26** was the only compound to alter the cell cycle in JURKAT cells. By contrast, it induced a little and statistically insignificant increase in mortality in healthy lymphocytes, without altering their cell cycle. This finding suggests that compound **26** may possess a specific anticancer activity. Although further investigations are needed to evaluate the effects of these compounds on non-tumorigenic cells, the preliminary data indicate a potential cell specificity, which could support their development as selective anticancer agents.

In conclusion, these preliminary biological assays support our assumption that quinoxaline-1,3,4-oxadiazole hybrid structures exhibit antitumor activity. However, further studies are ongoing to evaluate the impact of additional modifications, aiming to gather new information for the structure-based development of novel molecules.

## Data Availability

The original contributions presented in this study are included in the article/Appendix A. Further inquiries can be directed to the corresponding author(s).

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
