# Peer review of "Novel Oxadiazole-Quinoxalines as Hybrid Scaffolds with Antitumor Activity"

_ijms, 2025, doi:10.3390/ijms26041439_

Round 1
Reviewer 1 Report
Comments and Suggestions for Authors
In general, the use of English language is correct. However, a careful check is required.
A comment should be added in the chemistry part, concerning the 7-chloroderivative 37, that was the only one which was not used to the next reaction to prepare the corresponding carbohydrazide. Is there a stability issue?
In lines 89-91, the unambiguous structure determination of each pair of isomers should be described with more clarity. Supplementary material was not included in the provided files.
The chemistry part is well described, and the new compounds are well characterized. A possible addition that the authors could include, is to mention the recrystallization solvents for the new compounds.
The authors should rephrase lines 471-476 (I would suggest omitting “that” in line 474).
The Biological part should be significantly reduced. Details concerning the preliminary NCI screening tests, especially the single dose assays, must be avoided. The presentation of the corresponding data is only tiring. thus the text not attractive to the reader. The results should be presented briefly, with clarity. Structure-activity relationship studies are better extracted based on IC50, GI50 or related, statistically evaluated data (i.e. data presented in Table 7). For cytotoxicity tests a reference compound is usually added, for comparison reasons. On the other hand, a more precise comparison of the cytotoxicity against cancer cell lines over the cytotoxicity presented against human lymphocytes could be interesting, in order to evaluate the potential therapeutic window of the compounds.
Comments on the Quality of English LanguageIn general, the use of English language is correct. However, a careful check is required.
Author Response
Comment 1: A comment should be added in the chemistry part, concerning the 7-chloroderivative 37, that was the only one which was not used to the next reaction to prepare the corresponding carbohydrazide. Is there a stability issue?
Response 1: A comment has been added in page 4, line 106. The isomers 36 and 37 were isolated by FC as 34/35 and 38/39, but unlike other pairs, only derivative 36 was obtained in good yields, whereas isomer 37 in very low quantities, making no cost-effective its synthesis in large quantities.
Comment 2: In lines 89-91, the unambiguous structure determination of each pair of isomers should be described with more clarity. Supplementary material was not included in the provided files.
Response 2: The sentence in lines 89-91 (now 118-122, page 5) has been rewritten. Moreover, the file Supplementary material now has been uploaded.
Comment 3: The chemistry part is well described, and the new compounds are well characterized. A possible addition that the authors could include, is to mention the recrystallization solvents for the new compounds.
Response 3: Thank you for your comment. All final compounds were obtained by precipitation in ice/water of the hot POCl3 solution and washing with water the resulting precipitates. Fortunately, their purity, confirmed by 1H- and 13C-NMR spectra, and LC/MS analyses, did not require recrystallization of the final compounds.
Comment 4: The authors should rephrase lines 471-476 (I would suggest omitting “that” in line 474).
Response 4: Thank you for the comment. The sentence has been rewritten (now lines 433-438, page 19 in 3. Conclusions)
Comment 5: The Biological part should be significantly reduced. Details concerning the preliminary NCI screening tests, especially the single dose assays, must be avoided. The presentation of the corresponding data is only tiring. thus the text not attractive to the reader. The results should be presented briefly, with clarity.
Structure-activity relationship studies are better extracted based on IC50, GI50 or related, statistically evaluated data (i.e. data presented in Table 7).
For cytotoxicity tests a reference compound is usually added, for comparison reasons. On the other hand, a more precise comparison of the cytotoxicity against cancer cell lines over the cytotoxicity presented against human lymphocytes could be interesting, in order to evaluate the potential therapeutic window of the compounds.
Response 5: Thank you for your comment. According to your suggestion and referee 2, we fully revised this section making the text easier (pages 6-8 in red) and revising Table 4 by entering GI50 values of compounds 24, 25 and 26. Figures 2, 3 and 4 have been placed in Supplementary material as Fig. S57 – S59.
In response to your suggestion, we have clarified in the manuscript that for both cell death and cell cycle analysis, cells treated with DMSO (used to resuspend the compounds) served as the negative control, while cells treated with 5-fluorouracil were used as the positive control. Please refer to lines 340-341, page 16, of the revised manuscript for this information.
Regarding the comparison of cytotoxicity between tumor cell lines and human lymphocytes, only preliminary observations can be made at this stage. We compared flow cytometry data on cytotoxicity and cell cycle, highlighting the different effects of the compounds on tumor cells and healthy lymphocytes. Please refer to lines 475-485, page 20, of the revised manuscript.
4. Response to Comments on the Quality of English Language
Point 1: In general, the use of English language is correct. However, a careful check is required.
Response 1: Thank you for pointing this out. We carefully checked the language in the whole manuscript and corrections have been reported in red.
5. Additional clarifications
None

Reviewer 2 Report
Comments and Suggestions for Authors
The nitrogen containing heterocycles 1,3,4-oxadiazoles and quinoxalines are identified by the authors as possessing several structural features which are potentially useful for inclusion in the design of potential anticancer agents. In this paper, the authors report the synthesis and preliminary structure–activity relationship studies of a series of 25 new hybrid quinoxaline-1,3,4-oxadiazole derivatives, which are designed to contain different substituents both on the phenyl ring and on the quinoxaline heterocycle. Based on their previous work, the authors outline that their objective of this hybrid pharmacophore approach is to improve the antitumour activity of these heterocycles, and overcome potential problems of low solubility, multidrug resistance and adverse effects.
The synthetic routes to the target compounds are clearly described; the novel heterocyclic hybrids are spectroscopically characterised. A comprehensive series of hybrid heterocycles with diverse substitution on the aryl ring of the final products 4-28 are prepared.
The products 4-28 were initially evaluated in NCI 60 cell line human tumour screen and displayed interesting antiproliferative activity in many of the cell lines tested; compounds 24, 25 and 26 were particularly effective in breast cancer cell lines. These compounds (24, 25 and 26) were progressed to the 5 dose screening protocol in the NCI 60 cell line human tumour screen; GI50 values in the micromolar concentration range were obtained in many of the cell lines evaluated e.g. breast and leukaemia. A detailed SAR discussion is presented.
The cytotoxicity of 25 selected compounds was further investigated against additional human tumour cell lines NB4 (acute promyelocytic leukemia) and JURKAT (human T lymphocytic), SH-SY5Y (Neuroblastoma)and MAHLAVU (hepatocellular carcinoma), A375 (malignant melanoma) cell lines using a colorimetric test (MTS) with 5-fluorouracil (5-FU), 10 μM and 100 μM, as a positive control. The effect of the compounds on human lymphocytes as a model of non-tumour cells was also determined. The effect of compounds 24, 25 and 26 on cell cycle was determined in JURKAT and SH-SY5Y cells;
The compounds were shown to induce apoptosis or necrosis and to alter cell cycle. This effect was also present on human lymphocyte cultures. Some selectivity was demonstrated. The results are consistent with the induction of classical apoptosis, although necrosis or senescence are also possible mechanisms of cell death.
This work has identified a series of hybrid quinoxaline-1,3,4-oxadiazole derivatives, and demonstrated antiproliferative effects for the compounds; apoptosis is suggested as a possible mechanism of action for the compounds.
The following issues must be addressed by the authors before publication is considered.
1. Authors must provide more detailed background information on the choice of hybrid compounds and specifically compounds composed of hybrids of the nitrogen heterocycles 1,3,4-oxadiazoles and quinoxalines as the structures to be investigated in the present study.
2. A more detailed discussion of the chemistry illustrated in Scheme 1 and Scheme 2 is required:
3. Scheme 2: Authors must revise the illustration of the alkene for compounds 69 and 70 to indicate E and Z isomers.
4. Figures 2, 3 and 4 (NCI five dose assays of compounds 24, 25 and 26) should be removed, and placed in the Supplementary Information. The results for the GI50 value for each cell line for each compound 24, 25 and 26 should be combined in a single Table and included in the main paper.
5. It is not necessary to present both the Table and Figure to illustrate the results for the MTS cell viability assays for each cell line tested, (see Figures 5, 6, 7, 8, 9, 10, 11 and Tables 7, 8, 9, 10, 11, 12 and 13); Please use the graphical representation of results for the cell line in the text; the Tables can be included in the Supplementary Information.
6. Page 20; Figure 9 and Table 11 refer to cell line A375; however, there is not any reference in the text to cell line A375; please insert appropriate text.
7. Authors must revise and edit the text for the Biology section 2.2 for the one-dose screen results, (Pages 5, 6, 7, 8) as the text is excessively detailed in the description of the cytotoxicity activity of the compounds.
8. Based on the flow cytometry results obtained, apoptosis is suggested as a possible mechanism of action for the compounds. Have the authors performed any additional experiments to support this mechanism of action?
9. Authors indicate that the objective of the study in designing the hybrid compounds is “to overcome potential problems of low solubility, multidrug resistance and adverse effects” (See page 2); Authors must indicate how these objectives are addressed in the present work.
10. Authors must indicate any specific advantages of the compounds 24, 25 and 26 when compared with the control drug 7-FU in the cell viability assays on the selected cell lines
Author Response
Comment 1: Authors must provide more detailed background information on the choice of hybrid compounds and specifically compounds composed of hybrids of the nitrogen heterocycles 1,3,4-oxadiazoles and quinoxalines as the structures to be investigated in the present study.
Response 1: Thanks for your comment. We revised (in red) the introduction especially as regards these hybrid compounds (lines 76-89, pages 3 and 4).
Comment 2: A more detailed discussion of the chemistry illustrated in Scheme 1 and Scheme 2 is required:
Response 2: Agree. Integrations have been reported in red in the text (pages 4 and 5).
Comment 3: Scheme 2: Authors must revise the illustration of the alkene for compounds 69 and 70 to indicate E and Z isomers.
Response 3: Thank you for your comment, however, for graphical representations, crossed double bonds (or double either bonds) are a standard way to represent an undefined double bond and/or a mixture of isomers. In the specific case of compound 69, according to Yavari, I. et al. Synlett 12 (2009) 1921-1922 (doi:10.1055/s-0029-1217542) (reference 21 in the manuscript), as reported in lines 124 – 128, page 5, we obtained a mixture of four isomers: the pairs endo/exo and cis/trans isomers. The same result was obtained in the next synthetic step to 70. Therefore, in order not to overburden the scheme, we decided to use crossed double bonds both for structure 69 and 70 to indicate the eight isomers, instead of inserting four structures for compound 69 (isomers endo, exo, cis and trans) and another four isomers for compound 70.
Comment 4: Figures 2, 3 and 4 (NCI five dose assays of compounds 24, 25 and 26) should be removed, and placed in the Supplementary Information. The results for the GI50 value for each cell line for each compound 24, 25 and 26 should be combined in a single Table and included in the main paper.
Response 4: Done. Figures 2, 3 and 4 have been placed in Supplementary material as Fig. S57 – S59. Table 4 was revised by entering GI50 values of compounds 24, 25 and 26.
Comment 5: It is not necessary to present both the Table and Figure to illustrate the results for the MTS cell viability assays for each cell line tested, (see Figures 5, 6, 7, 8, 9, 10, 11 and Tables 7, 8, 9, 10, 11, 12 and 13); Please use the graphical representation of results for the cell line in the text; the Tables can be included in the Supplementary Information.
Response 5: Thank you for your comment. Tables 7, 8, 9, 10, 11, 12 and 13 have been placed in Supplementary material as Table S1-S7.
Comment 6: Page 20; Figure 9 and Table 11 refer to cell line A375; however, there is not any reference in the text to cell line A375; please insert appropriate text.
Response 6: Thank you for your comment, however, the A375 cell line had already been reported in manuscript (line 322, page 19, now line 301, page 13 in the revised version)
Comment 7: Authors must revise and edit the text for the Biology section 2.2 for the one-dose screen results, (Pages 5, 6, 7, 8) as the text is excessively detailed in the description of the cytotoxicity activity of the compounds.
Response 7: Thank you for your comment. According to your suggestion and referee 1, we fully revised this section making the text easier (pages 6 - 8 in red).
|
Comment 8: Based on the flow cytometry results obtained, apoptosis is suggested as a possible mechanism of action for the compounds. Have the authors performed any additional experiments to support this mechanism of action? Response 8: Thank you for your insightful suggestion. In this study, we assessed cell death through flow cytometry to distinguish between early apoptotic and necrotic cells. However, these preliminary results will require further validation using additional techniques, such as Western blotting (to evaluate PARP and Caspase activation), to reinforce our conclusions and provide a more thorough understanding of the apoptotic pathways triggered by our compounds. While the detailed analysis of apoptotic mechanisms is not the primary focus of this study, ongoing research aims to expand on these biological investigations. We intend to publish our findings in a forthcoming publication.
Comment 9: Authors indicate that the objective of the study in designing the hybrid compounds is “to overcome potential problems of low solubility, multidrug resistance and adverse effects” (See page 2); Authors must indicate how these objectives are addressed in the present work. Response 9: Out of several measures adopted to counter the drawbacks of multidrug resistance, side effects, hybrid drugs offer a promising strategy. Therefore, due to our experience in 1,3,4-oxadiazole and quinoxaline scaffolds as antitumor agents, we hypothesized that their hybrids may have anti-tumor activity as confirmed by preliminary assays presented in the paper. Concerning the hypothesis on multidrug resistance and adverse effects, these data can only be obtained by advanced pre-clinical studies, which we have not conducted. |
However, taking into account your comment, if upcoming SARs on the most interesting compounds 24-26 will allow us to identify some particularly interesting derivatives, further biological studies will be conducted.
|
Comment 10: Authors must indicate any specific advantages of the compounds 24, 25 and 26 when compared with the control drug 5-FU in the cell viability assays on the selected cell lines. |
Response 10: Thank you for your suggestion. We have discussed the potential advantages of compounds 24, 25, and 26 in comparison to the control drug 5-fluorouracil, with a focus on their effects on cytotoxicity and cell cycle alterations. Please refer to lines 459-479 of the revised manuscript for further details.
Response to Comments on the Quality of English Language
Point 1:
Response 1: We carefully checked the language in the whole manuscript and corrections have been reported in red.
Additional clarifications
None

Reviewer 3 Report
Comments and Suggestions for Authors
The manuscript focuses on the synthesis of 25 novel 1,3,4-oxadiazole-quinoxaline derivatives, followed by their in vitro biological evaluation on nine National Cancer Institute tumour cell lines to test their cytotoxic activity. Compounds 24, 25 and 26 were the most promising, showing strong activity in various cell lines, particularly against breast cancer and leukaemia. Among these, compound 24 showed a GI50 of 1.85 μM against the MCF7 line, being the most effective of the entire library tested. In addition, a structure-activity analysis (SAR) was performed to identify the relationship between the chemical structure of the compounds and their anti-cancer activity. It was observed that the presence of a trifluoro-methylated group in the C-7 position of the quinoxaline moiety is critical to the efficacy of compounds, while other changes in structure, such as the introduction of groups like methoxyl and chlorine, have variable effects on activity. Studies on the mechanism of action indicated that the compounds induce apoptosis in cancer cells, with significant alterations in the cell cycle, particularly in S-phase. Cytotoxicity on human lymphocytes is lower, suggesting some selectivity for cancer cells.
The manuscript is well structured, and I will recommend the acceptance after minor revisions (listed below).
- The authors should consider citing this work (DOI: 10.1021/acschembio.0c00039), which highlights the antitumoral activity of pyridoxal phosphate (PLP) and inosine monophosphate (IMP). These natural heterocyclic compounds have demonstrated significant antitumor potential, particularly against chronic myeloid leukemia (CML). The inclusion of this reference would enhance the discussion on the importance of nitrogen-based heterocycles in cancer drug discovery, further highlighting their role as stable and effective scaffolds for targeting specific mechanisms of action in cancer therapy.
- The authors comprehensively reported GI50 values of the most promising compounds using breast cancer cell lines (e.g. MCF7 and MDA-MB-468) and leukemic cell lines (e.g. CCRF-CEM, HL-60(TB)). The authors could add a brief discussion as to why these cell lines are particularly reactive. Is it related to specific molecular pathways?
- Authors should correct minor typographical errors (e.g. replace, ‘able of interact’ in line 31 with ‘able to interact’).
Author Response
Comments 1: The authors should consider citing this work (DOI: 10.1021/acschembio.0c00039), which highlights the antitumoral activity of pyridoxal phosphate (PLP) and inosine monophosphate (IMP). These natural heterocyclic compounds have demonstrated significant antitumor potential, particularly against chronic myeloid leukemia (CML). The inclusion of this reference would enhance the discussion on the importance of nitrogen-based heterocycles in cancer drug discovery, further highlighting their role as stable and effective scaffolds for targeting specific mechanisms of action in cancer therapy.
Response 1: Thank you for the comment. Reference [2] has been added in the manuscript
Comments 2: - The authors comprehensively reported GI50 values of the most promising compounds using breast cancer cell lines (e.g. MCF7 and MDA-MB-468) and leukemic cell lines (e.g. CCRF-CEM, HL-60(TB)). The authors could add a brief discussion as to why these cell lines are particularly reactive. Is it related to specific molecular pathways?
Response 2: Thank you for the comment. Further studies to deepening biological investigations, i.e. specific target are ongoing and will be published in a forthcoming publication.
Response to Comments on the Quality of English Language
Point 1: Authors should correct minor typographical errors (e.g. replace, ‘able of interact’ in line 31 with ‘able to interact’).
Response 1: Thank you for pointing this out. We carefully checked the language in the whole manuscript and corrections have been reported in red.
Additional clarifications
None

Round 2
Reviewer 1 Report
Comments and Suggestions for Authors
The authors have addressed most of the comments, and the manuscript can now be accepted for publication.
Reviewer 2 Report
Comments and Suggestions for Authors
Authors have completed the revisions requested; the manuscript can be accepted